# Higher-order assemblies of oligomeric cargo receptor complexes form the membrane scaffold of the Cvt vesicle

Chiara Bertipaglia[1,†], Sarah Schneider[1,†], Arjen J Jakobi[1,2], Abul K Tarafder[1], Yury S Bykov[1], Andrea Picco[3,‡], Wanda Kukulski[1,3,§], Jan Kosinski[1], Wim JH Hagen[1], Arvind C Ravichandran[1], Matthias Wilmanns[2], Marko Kaksonen[3,‡], John AG Briggs[1,3] & Carsten Sachse[1,*]

## Abstract

Selective autophagy is the mechanism by which large cargos are specifically sequestered for degradation. The structural details of cargo and receptor assembly giving rise to autophagic vesicles remain to be elucidated. We utilize the yeast cytoplasm-to-vacuole targeting (Cvt) pathway, a prototype of selective autophagy, together with a multi-scale analysis approach to study the molecular structure of Cvt vesicles. We report the oligomeric nature of the major Cvt cargo Ape1 with a combined 2.8 Å X-ray and negative stain EM structure, as well as the secondary cargo Ams1 with a 6.3 Å cryo-EM structure. We show that the major dodecameric cargo prApe1 exhibits a tendency to form higher-order chain structures that are broken upon interaction with the receptor Atg19 *in vitro*. The stoichiometry of these cargo–receptor complexes is key to maintaining the size of the Cvt aggregate *in vivo*. Using correlative light and electron microscopy, we further visualize key stages of Cvt vesicle biogenesis. Our findings suggest that Atg19 interaction limits Ape1 aggregate size while serving as a vehicle for vacuolar delivery of tetrameric Ams1.

**Keywords** autophagy cargo; autophagy receptor; correlative light and electron microscopy; electron cryomicroscopy; fluorescence light microscopy; selective autophagy; X-ray crystallography
**Subject Categories** Autophagy & Cell Death; Membrane & Intracellular Transport

## Introduction

Macroautophagy (hereafter referred to as autophagy) is a highly conserved process of intracellular clearance in eukaryotes through which cytosolic components are engulfed into double-membrane vesicles and transported to degradative compartments. During this process, autophagosomes encapsulate large molecular cargos such as macromolecules, protein aggregates, organelles, viruses, and bacteria that are too large in size to be degraded by the ubiquitin proteasome system [1]. The process is thought to be mediated by a total of 41 proteins [2] and commences with the formation of a cup-shaped phagophore in the cytosol triggered by the autophagy core machinery [3]. This machinery includes the Atg1 kinase, the class III phosphatidylinositol 3-kinase, and the Atg12–Atg5–Atg16 complex that finally conjugates the C-terminus of Atg8 to phosphatidylethanolamine (PE) lipids. Thus, the autophagy core machinery decorates the expanding phagophore with Atg8-PE and mediates the growth of the double-membrane vesicle. In the past years, it has emerged that many cargos are recognized in a highly selective fashion by specific receptors that function as molecular adaptors bridging the cargo with the autophagosomal marker Atg8/LC3 on forming autophagosomes, [1,4–7].

The earliest discovered selective autophagy pathway in *Saccharomyces* (*S.*) *cerevisiae* is the cytoplasm-to-vacuole targeting (Cvt) pathway, through which hydrolytic enzymes are sequestered into double-membrane vesicles, termed Cvt vesicles, and transported to the vacuole [8,9]. Despite being a biosynthetic pathway, the Cvt pathway utilizes the core autophagy machinery through adaptors that bind the Atg1 complex and Atg8 [10,11]. Once the Cvt vesicle outer membrane fuses with the vacuole, the inner membrane of the remaining Cvt body [12,13] is degraded and the enzymes are released into the vacuolar lumen, where the proteases are subjected

1  Structural and Computational Biology Unit, European Molecular Biology Laboratory, Heidelberg, Germany
2  Hamburg Unit, European Molecular Biology Laboratory, Hamburg, Germany
3  Cell Biology and Biophysics Unit, European Molecular Biology Laboratory, Heidelberg, Germany
   *Corresponding author. Tel: +49 6221 3878407; Fax: +49 6221 3878519; E-mail: carsten.sachse@embl.de
   †These authors contributed equally to this work
   ‡Present address:  Department of Biochemistry, University of Geneva, Geneva, Switzerland
   §Present address:  Cell Biology, MRC Laboratory of Molecular Biology, Cambridge, UK
   The copyright line of this article was changed on 25 August 2016 after original online publication

to a process of maturation [12,14,15] and finally are catalytically active to degrade vacuolar components.

The best-characterized cargo component of the Cvt vesicle is the vacuolar aminopeptidase I (Ape1) [16]. Ape1 is synthesized in the cytosol as a 55-kDa zymogen that includes an N-terminal propeptide (hereafter referred to as prApe1), which oligomerizes into a dodecamer [17,18] and further assembles into larger aggregates in a propeptide-dependent manner [19]. The resulting prApe1 complex is considered an indispensible cargo of the Cvt vesicle and is recognized by the selective Atg19 receptor via the N-terminal propeptide [19–24]. Following fusion of the Cvt vesicle with the vacuole, cargo is released into the vacuolar lumen where prApe1 undergoes processing to mature Ape1 (hereafter referred to as mApe1) by cleavage of the N-terminal propeptide. In addition to prApe1, Atg19 also binds another cargo hydrolase α-mannosidase 1 (Ams1) [25,26] via a site distinct from the prApe1 binding site, thus forming the Cvt complex. In contrast to the major cargo Ape1, Ams1 was shown to be dispensable and is considered a secondary cargo of the Cvt vesicle [23]. The Cvt complex interacts with Atg11 via binding of phosphorylated Atg19 [27] and travels toward the pre-autophagosomal structure where Atg19 binds lipidated Atg8 to associate the Cvt complex with the autophagy core machinery and the nascent Cvt vesicle [23].

As the Cvt pathway utilizes a large proportion of the yeast autophagy protein machinery, it is considered a prototype system to study the structure and dynamics of selective autophagy in eukaryotes [28]. Despite the fact that the Cvt vesicle carries a relatively small number of well-defined cargo proteins that are recognized by the autophagy machinery, the assembly structures of the cargos and the interplay with the autophagy receptor Atg19 are poorly understood. Moreover, the stoichiometry and how they give rise to the observed ultrastructure of the Cvt vesicle in the cell remain to be established. Here, we have elucidated the X-ray and cryo-EM structures of the major cargo Ape1 and the secondary cargo Ams1, respectively. We have also investigated the oligomeric state of the receptor Atg19, the propensity of the major cargo to form higher-order assemblies, and how they interact with Atg19. We determined the stoichiometry of these components *in vivo* and show that the relative levels of Atg19 and Ape1 are key to maintaining the size of Cvt structures. Further, we have visualized key stages of Cvt vesicle biogenesis in the cellular environment using correlative light and electron microcopy (CLEM). Our results allow the construction of a structural model of the Cvt vesicle at multiple levels of resolution as it occurs in the cellular context. The data also reveal how molecular assemblies of cargo and receptor proteins serve as important delivery vehicles and scaffolds that guide the formation of selective autophagy vesicles.

# Results

### prApe1 dodecamers form higher-order assembly structures

In order to shed light on the molecular organization of the major cargo of the Cvt vesicle, we determined the 3D structural model of Ape1 using a combined electron microscopy and X-ray crystallography approach. First, we expressed full-length *S. cerevisiae* prApe1 and imaged the protein using negative stain electron

microscopy (EM) (Fig 1A). We observed particles of 18 nm diameter including views compatible with described dodecamers of mApe1 [17,18]. Interestingly, a large proportion of them are capable of forming connected doublets, triplets, and larger assemblies hampering further more detailed structural analysis. When expressing *S. cerevisiae* mApe1 (sc-mApe1) lacking the N-terminal propeptide, we predominantly observed isolated dodecamers when imaged at identical protein concentration and buffer conditions (Fig 1A–C), indicating that the propeptide stimulates the *trans*-interactions between multiple prApe1 dodecamers. Due to the presence of smaller particles of putative mApe1 hexamers (Fig 1B, white arrowheads), we stabilized and purified sc-mApe1 using a GraFix gradient [29] (Fig 1D). Based on the twofold and threefold end-on views from class average analysis (Fig 1E), we determined a low-resolution negative stain structure at 24 Å resolution by imposing tetrahedral symmetry (Figs 1F and G, and EV1E). In order to further improve the resolution of the mApe1 EM structure, we used an Ape1 homolog of 40% sequence identity from the thermophilic filamentous fungus *Chaetomium* (*C.*) *thermophilum* (ct-mApe1) [30], to exploit increased stability of dodecameric species when compared with sc-mApe1 (Fig EV1). Moreover, we determined the crystal structure of ct-mApe1 organized in dodecamers at 2.8 Å resolution. In the crystal, ct-mApe1 is present as dodecamers, which are formed by two asymmetric units each containing six ct-mApe1 molecules (Table 1, Materials and Methods). The ct-mApe1 X-ray structure is compatible with the dimensions of the sc-mApe1 EM projections and fits very well into the tetrahedral low-resolution EM density (Figs 1G and EV2). The crystal structure of the ct-mApe1 dodecamer has an RMSD of 0.7 Å on Cα atoms in comparison with the previously determined crystal structure of sc-mApe1 (PDB 4r8f) [31] (Fig EV2G and H). The superposition shows a close structural overlap of the dodecamer assemblies except for a noticeable difference in the active site that is well defined in our structure due to the presence of the cofactor zinc, but disordered in the sc-mApe1 crystal (Fig EV2G). Moreover, the geometric positioning of the N-terminal propeptide on the surface of the dodecamer is consistent with our observed *trans*-interactions of dodecamers leading to higher-order assemblies (Fig 1A, C and G). In contrast to previous studies that analyzed detergent-solubilized prApe1 preparations [31], we show that untreated *in vitro* purified prApe1 has the tendency to form larger assemblies and may thus contribute to a higher level of structural organization in addition to the described oligomeric state of dodecamers.

### Cryo-EM structure of tetrameric *Saccharomyces cerevisiae* Ams1

In order to further the structural analysis of the components in Cvt vesicles, we determined the 3D structure of the secondary cargo Ams1 from *S. cerevisiae* using cryo-EM. After expression and purification of 125-kDa full-length Ams1, we imaged negative stain embedded samples using EM and observed homogeneous particles of 13 × 16 nm in dimension (Fig 2A). The images revealed class average views with two perpendicular twofold rotation axes indicative of dihedral symmetry (Fig 2A) and were used for determination of an initial low-resolution 3D model of Ams1. After vitrification of the sample (Fig 2B and C), we refined the tetrameric Ams1 structure to 6.3 Å overall resolution from 33,588 particles (Figs 2D and E, and EV3A). Subsequently, we built a complete pseudo-atomic model

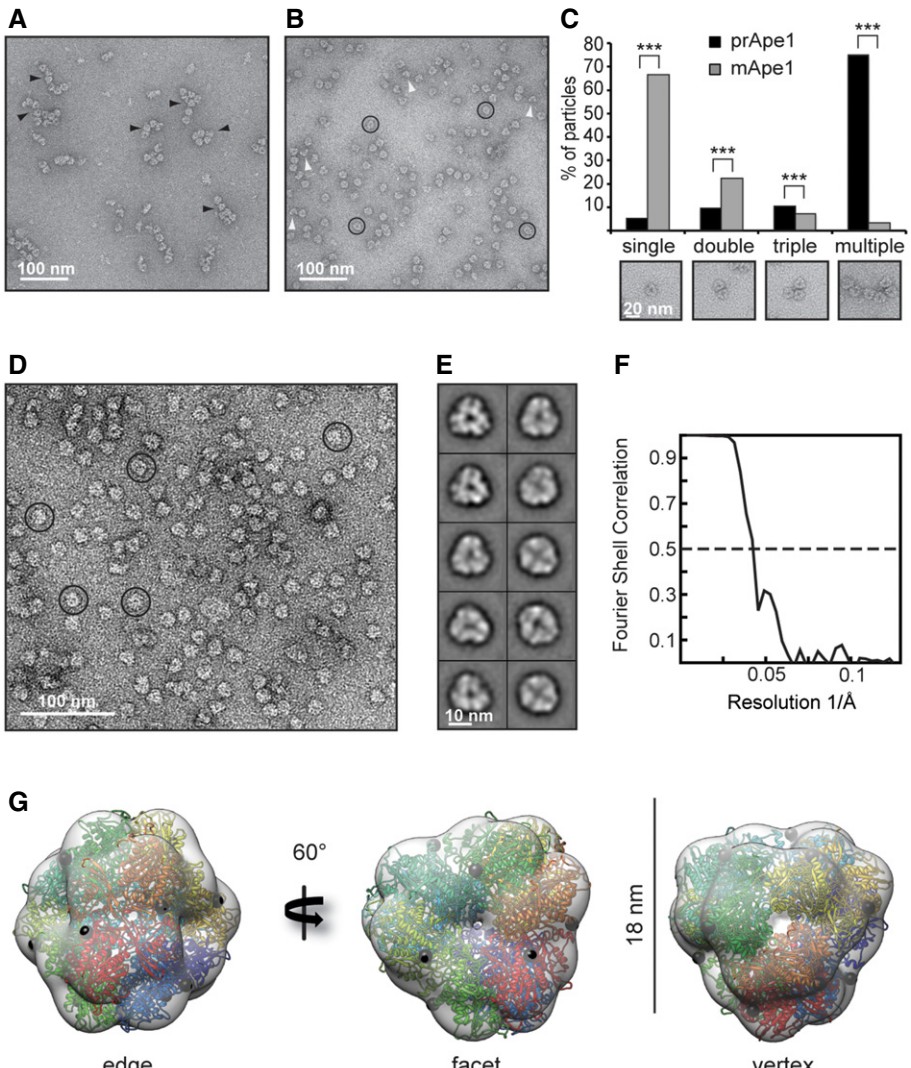

**Figure 1.  *Saccharomyces cerevisiae* prApe1 higher-order organization and *Chaetomium thermophilum* mApe1 crystal structure fitted into *S. cerevisiae* mApe1 low-resolution EM map.**

A   Negative stain electron micrograph showing the tendency of *S. cerevisiae* prApe1 (sc-prApe1) dodecamers to form higher-order assemblies or chains (black arrowheads).

B   Negative stain electron micrograph of *S. cerevisiae* mApe1 (sc-mApe1) showing monodisperse dodecameric particles (circles) as well as hexameric particles (white arrowheads).

C   Bar plot showing the occurrence of single, double, triple, or multiple aggregates of sc-prApe1 in comparison with sc-mApe1. Insets show single, double, triple, or multiple dodecamers. A total of $n$ = 3,186 particles were counted in the case of sc-prApe1 and $n$ = 3,107 particles were counted in the case of sc-mApe1. Particles were picked from 20 different micrographs. ***$P < 2.2 \times 10^{-16}$ (Pearson's chi-squared test).

D   Negative stain electron micrograph of GraFixed sc-mApe1 showing dodecameric particles (circles).

E   Class averages of sc-mApe1 dodecamers including views of twofold and threefold symmetry.

F   Fourier shell correlation of the EM reconstruction of the sc-mApe1 dodecamer indicates a resolution of 24 Å.

G   *C. thermophilum* mApe1 crystal structure fitted into electron microscopy (EM) map of sc-mApe1 dodecamer. Each of the 12 chains is depicted by a different color. A black sphere represents the N-terminus where the propeptide would emanate from the dodecamer. From left to right: twofold axis view from the edge, threefold axis view from the facet, and threefold axis view from the vertex.

based on a homology model of the mannosidase core combined with two additional N-terminal domains (Fig 2F). The mannosidase core (residues 287–1083) was derived from *Streptoccocus* (*S.*) *pyogenes* α-mannosidase (PDB 2wyh) that contains three domains present in Ams1: an α/β barrel, a three-helix bundle, both of which participate in the catalytic center (Fig EV3B), as well as a

β-sandwich domain (Fig 2G and H). For the remaining N-terminus, we built two domains: a jelly-roll fold (residues 45–203) and a four-helix bundle (residues 17–27, 209–271) based on distant homology and the EM density (Movie EV1).

In contrast to the homologous dimeric *S. pyogenes* α-mannosidase [32], *S. cerevisiae* Ams1 forms a tetramer. The transverse

**Table 1.    Data collection and refinement statistics.**

| Data collection | |
| --- | --- |
| Resolution range (Å) | 58.7–2.75 (2.8–2.75) |
| Space group | P 2 2$_1$ 2$_1$ |
| Cell dimensions | |
| *a, b, c* (Å) | 121.0, 143.9, 201.3 |
| $\alpha = \beta = \gamma$ (°) | 90 |
| $R_{merge}$ | 0.13 (0.77) |
| CC$_{0.5}$ | 0.98 (0.52) |
| $<I>/s(I)$ | 7.7 (1.3) |
| Completeness (%) | 98.7 (93.5) |
| Multiplicity | 3.1 (3.0) |
| **Refinement** | |
| Resolution (Å) | 49.1–2.75 |
| No. reflections | 172,990 |
| $R_{work}/R_{free}$ (%) | 19.2 (30.5)/23.9 (36.2) |
| No. atoms | 20,614 |
| Protein | 20,394 |
| Ion | 12 |
| Water | 208 |
| B-factors (Å$^2$) | |
| Protein | 24.6 |
| Ion | 26.8 |
| Water | 47.3 |
| R.m.s deviations | |
| Bonds (Å) | 0.012 |
| Angles (°) | 1.526 |

interface connecting two monomers perpendicularly to the Ams1 length axis is formed by the β-sandwich domain and is located on the opposite side of the dimerization interface described in the homologous mannosidase structure (Fig 2I and J, and Movie EV1). In addition, we identified an N-terminal four-helix bundle domain

that comprises the major longitudinal interface bridging two monomers along the Ams1 length axis within the tetramer (Fig 2I and K). To validate whether the surface formed by the four-helix bundle is responsible for mediating longitudinal contacts within the tetramer, we generated a W234E mutant by introducing a negatively charged residue to selectively disrupt this longitudinal interface (Fig 2K). While the Ams1 W234E mutant is still catalytically active (Fig EV3C), size-exclusion chromatograms show a peak shift toward smaller molecular species in comparison with wild-type Ams1 (Fig EV3D). When the fractions were analyzed by negative staining EM, we predominantly observed particles smaller than wild-type tetramers (Fig 2L). Classification of the W234E particles revealed smaller two-winged class averages of 13 × 10 nm dimensions corresponding to approximately half the size of tetrameric Ams1. Comparison with simulated reprojections of putative Ams1 dimers strongly suggests a dimeric Ams1 complex with longitudinal contacts disrupted and transverse contacts still maintained (Fig 2L). Moreover, in order to test whether these Ams1 samples have the capability of binding the native receptor Atg19, we subjected Ams1 wild-type tetramers and W234E dimers to an Atg19 pull-down assay. The corresponding SDS–PAGE shows that wild-type tetrameric Ams1 binds MBP-Atg19 efficiently, whereas the introduction of a W234E mutation drastically reduces binding (Fig EV3E). This suggests that the Ams1 wild-type tetramer is required for *in vitro* recognition by the autophagy receptor Atg19. Together, our cryo-EM data and the pseudo-atomic model reveal that the N-terminal four-helix bundle domain is critical for the Ams1 tetrameric assembly.

**Atg19 forms a trimer competing with prApe1 higher-order assembly formation**

Next, we focused on characterizing the higher-order molecular organization of the prApe1 cargo from *S. cerevisiae* as it is found in the Cvt complex together with Atg19. Atg19 is the selective receptor that recognizes the major cargo prApe1 and the secondary cargo Ams1 to form the Cvt complex [4,22,23]. We expressed and purified *S. cerevisiae* Atg19 to biochemically characterize the interactions with the major cargo Ape1 in more detail. Using size-exclusion chromatography coupled to

**Figure 2.    Cryo-EM structure of *Saccharomyces cerevisiae* Ams1.**

A   Electron micrograph of negatively stained Ams1 with particles and higher-order assemblies (arrows). Class averages exhibit two distinct twofold views consistent with D2 symmetry. Scale bar, 10 nm.
B   Electron cryomicrograph of Ams1 with particles highlighted (circles).
C   Selected particles (top row) are shown with corresponding class averages (bottom row). Scale bar, 10 nm.
D   Fourier shell correlation indicates a resolution of 6.3 Å according to the 0.143 criterion.
E   Top view of the cryo-EM structure, four Ams1 molecules were segmented and colored (dark blue, beige, red, light blue).
F   Molecular architecture of Ams1 monomer. Primary structure, two map segments with fitted pseudo-atomic models are shown. Left: The N-terminal portion was built from distant homology models: a four-helix bundle (17–27, 209–271) in yellow and a jelly-roll fold (45–203) in red. Right: The α-mannosidase core (287–1,083) with an α/β barrel (287–573) in blue, a three-helix bundle (574–671) in green, and the β-sandwich domain (672–1,083) in purple.
G   Density section through the Ams1 monomer with three-helix bundle in the center.
H   Complete atomic model of Ams1 with active site indicated as yellow sphere fitted into the cryo-EM map. Corresponding 3D structure is shown in Movie EV1.
I   Schematic representation of Ams1 domain distribution highlighting the molecular interfaces within the tetramer.
J   The β-sandwich domain (purple) makes up the transverse interface between chain A and chain B.
K   The N-terminal four-helix bundle (orange) mediates the longitudinal interface. Asterisk denotes position of W234E mutation between chains A and C/D.
L   Negatively stained W234E Ams1 mutant reveals predominantly smaller particles when compared with wild-type tetramers. Representative class averages show a two-lobed structure consistent with reprojections of chain A and B (dark blue and beige) dimers depicted in ribbon and surface representation below. Scale bar, 10 nm.

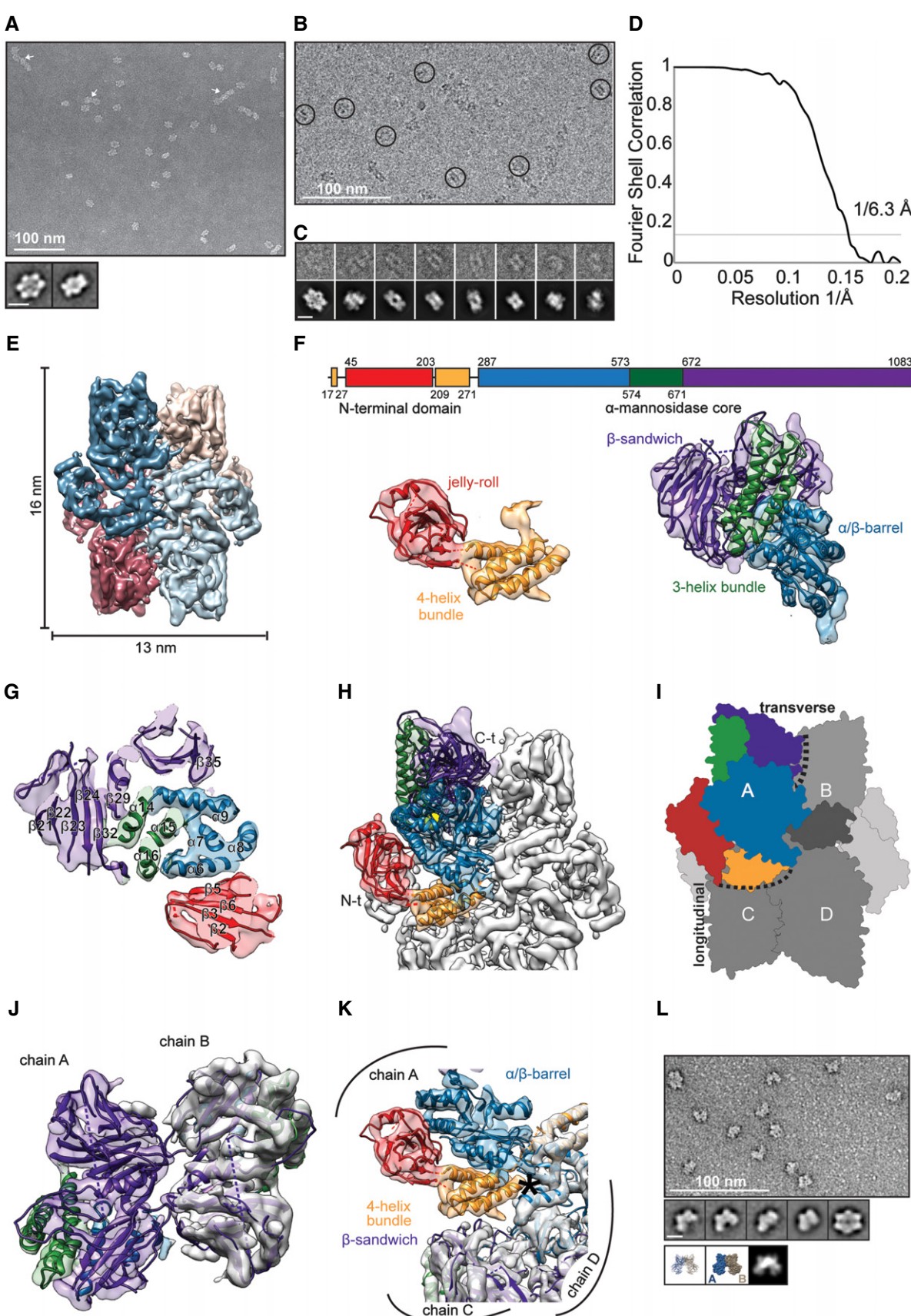

**Figure 2.**

multi-angle light scattering (SEC-MALS), we determined a molecular weight of 142 kDa corresponding to a trimeric homo-oligomer in solution (Fig 3A). By mixing purified prApe1 dodecamers together with Atg19, we reconstituted the Atg19/prApe1 complex and showed that Atg19 selectively binds prApe1 and not mApe1, confirming the importance of the propeptide for Atg19 interaction and demonstrating the binding to full-length prApe1 for the first time (Fig 3B). We further analyzed the cargo–receptor complex by co-expressing prApe1 and Atg19. The binary complex was purified by affinity chromatography (Fig 3C) and subjected to size-exclusion chromatography where Atg19 and prApe1 co-eluted as a complex larger than 500 kDa (Figs 3D and EV4A and B). Negative staining EM of the purified binary complex revealed two different types of particles, reminiscent either of prApe1 dodecamers alone or of dodecamers with extra density associated with the particle (Fig 3E and F). Class averages confirmed that 20% of the particles had extra density at the periphery of the tetrahedral envelope, indicative of Atg19 binding. Interestingly, prApe1 dodecamers that co-purified with Atg19 displayed a reduced tendency to form higher-order assemblies (Figs 1A and 3E and G). In line with this observation, prApe1 dodecamers became more soluble in the presence of Atg19 in pelletation assays, in comparison with prApe1 dodecamers alone (Fig EV4C and D). The observed change of prApe1 assembly state reveals that trimeric Atg19 competes with propeptide *trans*-interactions between different dodecamers, thereby solubilizing higher-order assemblies *in vitro* and having the potential to also regulate the relative size of the Cvt aggregate. Given our observation, we hypothesized that prApe1 forms larger structures in the absence of Atg19 *in vivo*. In order to test this hypothesis, we labeled Cvt cargo prApe1 with green fluorescent protein (GFP) in *S. cerevisiae* ypt7Δ strains, which led to enrichment of prApe1 puncta due to block of vesicle fusion with the vacuole [33]. Subsequently, we analyzed the intensity of prApe1-GFP spots in living cells, either in the presence or in the absence of Atg19 (Figs 3H and EV4E). In line with our hypothesis, we measured the prApe1 fluorescence intensity in the Atg19 knockout strain to be on average $2.3 \pm 0.6$ times brighter while the overall prApe1 level remained constant when compared with the Atg19 control strain (Figs 3I and EV4E). Together, our data show that the level of the autophagy receptor Atg19 controls the size of the prApe1 structures *in vitro* and *in vivo*.

## Stoichiometry of the Cvt complex determined by quantitative fluorescence microscopy *in vivo*

As the major cargo prApe1 is commonly found in large protein-dense aggregates of the cellular Cvt complex or vesicle [12], we wanted to characterize the stoichiometry of the Cvt complex in living yeast cells to assess how the observed tendency of forming size-regulated higher-order assemblies is determined by the relative concentrations of prApe1, Ams1, and Atg19 *in vivo*. To measure the stoichiometry of the prApe1/Atg19/Ams1 complex, we labeled the Cvt cargos prApe1 or Ams1 as well as the Atg19 receptor with GFP. For each of the GFP-labeled proteins, we then compared the relative fluorescence intensities in cells expressing Nuf2-GFP, whose molecular abundance had been calibrated previously [34–36]. As our study focuses on the Cvt pathway, we evaluated only prApe1-GFP or Ams1-GFP puncta colocalizing with Atg19-mCherry, thereby excluding free prApe1 or Ams1 complexes in the cytosol [4] (Fig 4A). Fluorescence quantification from these three yeast strains allowed us to determine the relative abundance of Cvt cargos and receptor, as $3,585 \pm 318$ molecules of prApe1, $501 \pm 62$ molecules of Ams1, and $332 \pm 35$ molecules of Atg19 per punctum were measured. Our results indicate that Ape1 is ~10 times more abundant than Atg19, whereas Ams1 is on average approximately 2 times more abundant than Atg19 (Fig 4B). The 7-fold excess of prApe1 over Ams1 confirms the categorization that prApe1 is the major and Ams1 the secondary cargo. The still relatively high ratio of autophagy receptor Atg19 suggests that instead of simply bridging decorated cargo assemblies with the membrane, Atg19 is also incorporated into Cvt aggregates, which is in line with the suggested role in the size regulation as indicated from our *in vitro* and *in vivo* interaction studies.

**Figure 3.  prApe1/Atg19 complex.**

A    SEC-MALS profile of Atg19 yields a molecular weight estimate of 142 kDa (measured in triplicate).

B    Right: *In vitro* pull-down binding assay and subsequent SDS–PAGE. Amylose beads are used as a bait to demonstrate interaction of purified prApe1 dodecamers with MBP-Atg19 (lane 1), whereas purified mApe1 dodecamers are not capable of interacting with MBP-Atg19 (lane 3). As controls, MBP does not bind prApe1 and mApe1 (lanes 2 and 4). Left: SDS–PAGE showing input of the corresponding pull-down assay.

C    Co-expression of MBP-Atg19 and prApe1 in Sf21 cells; SDS–PAGE of eluate from amylose beads.

D    Size-exclusion chromatography profile (Superose 6 column) of the prApe1/Atg19 complex after anti-MBP-Atg19 purification and cleavage of the MBP tag (top). SDS–PAGE of peak fractions of the gel filtration profile (bottom). Samples were run on two separate gels.

E    Negative stain electron micrograph corresponding to the size-exclusion chromatography peak at 10 ml retention volume of (D). Several prApe1 dodecamers possess extra density corresponding to Atg19 and are highlighted in the insets.

F    Comparison between class averages of purified prApe1 dodecamers with class averages of prApe1 dodecamers that were co-purified with Atg19.

G    Bar plot of occurrence of single, double, triple, and multiple dodecamers from the prApe1 and the prApe1/Atg19 sample. A total of $n = 3,186$ particles were counted in the case of prApe1 and $n = 3,340$ particles were counted in the case of prApe1/Atg19. Particles were picked from 20 different micrographs. ***$P < 2.2 \times 10^{-16}$ (Pearson's chi-squared test).

H    Epifluorescence microscopy images of the prApe1-GFP/Atg19-mCherry/ypt7Δ, prApe1-GFP/Atg19Δ/ypt7Δ *Saccharomyces cerevisiae* cells that were used to analyze the fluorescence intensity of all prApe1-GFP spots. BF represents bright-field images.

I    Bar plot showing the ratio between the GFP intensity of prApe1 spots in prApe1-GFP/Atg19-mCherry/ypt7Δ versus prApe1-GFP/Atg19Δ/ypt7Δ *S. cerevisiae* cells. The intensity of Ape1-GFP was normalized over the average fluorescence intensity of Nuf2-GFP measured in kinetochores as performed in [36]. A total of 54 prApe1-GFP spots and 80 Nuf2-GFP spots were analyzed in prApe1-GFP/Atg19Δ/ypt7Δ cells, whereas 52 prApe1-GFP spots and 98 Nuf2-GFP spots were analyzed in prApe1-GFP/Atg19-mCherry/ypt7Δ control cells. Error bars indicate standard error of the mean (SEM). *$P = 0.03$ (Z test).

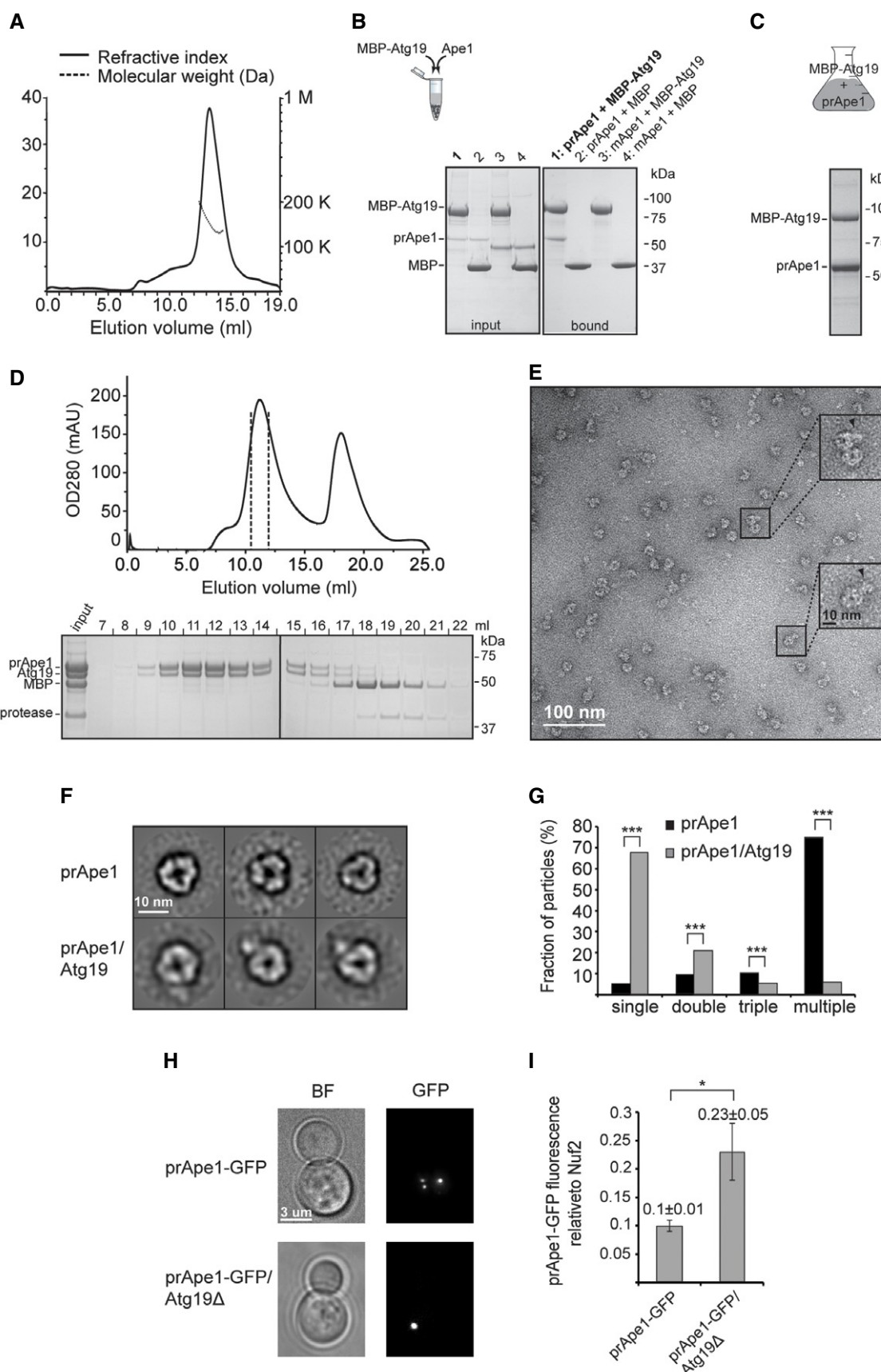

**Figure 3.**

## Correlative light and electron microscopy identifies multiple stages of Cvt vesicle biogenesis

Given our *in vitro* structural and biochemical findings on the major prApe1 and secondary Ams1 Cvt cargos, as well as the Atg19 receptor, we wanted to relate them with the *in vivo* Cvt vesicle ultrastructure. Therefore, we visualized and statistically evaluated their morphology and shapes using a recently described CLEM procedure coupled to electron tomography [37]. For this approach, we tagged prApe1 with GFP and Atg19 with mCherry in ypt7Δ *S. cerevisiae* cells (Fig 5A) to specifically localize Cvt vesicles in plastic sections by 3D electron tomography (Fig 5B and Movie EV2). In order to minimize the effect of labels, we also prepared and imaged sections from cells expressing solely prApe1-GFP (Fig EV5A–G). Qualitatively, we found that in both data sets, the morphological features appeared very similar (Fig EV5H–O). From a total of 38 correlated tomograms, 10 locations were poorly preserved and excluded from further analysis. From the remaining 28 sites, 26 cases displayed dense homogeneous protein content ranging in diameter from 120 to 250 nm (Fig 5C), compatible with the 150 nm reported in earlier EM studies [12] (Fig EV5A–F and H–N). We found that the average observed Cvt vesicle is an ellipsoid with a cross section of 165 × 185 nm dimensions, deviating from a perfect circular cross section of a sphere by an average ellipticity of 11%. Although we specifically targeted prApe1 for visualization, we did not recognize the molecular shapes of individual 18 nm wide prApe1 dodecamers due to the poor contrast and limited preservation after resin embedding. Interestingly, two tomograms showed autophagosomes that were larger than 300 nm in *x* and *y* dimensions and easily discernible from Cvt vesicles due to the presence of cytoplasmic content such as other membranes and ribosomes [12,22] (Fig EV5G and O). We categorized the well-defined correlated sites by grouping the tomograms into four distinct ultrastructures: cytoplasmic Cvt aggregate without membrane structures (29%) (Figs 5D and EV5N, and Movie EV3), partially membrane-enwrapped Cvt aggregate (10%) (Figs 5E and EV5F, and Movie EV4), double-membrane Cvt vesicles (54%) (Figs 5F and EV5A–E and H–M, and Movie EV5), and autophagosomes (7%) (Fig EV5G and O). Our data show that Atg19 is already present in cytoplasmic Cvt aggregates also in the absence of membrane structures. In a total of 20 sites (71%), we identified membrane structures from Cvt vesicles or autophagosomes, which were characterized by the presence of two juxtaposed bilayers of 11.8 ± 1.7 nm width that are approximately two times wider than the other cellular membranes present in the tomograms (Fig 5B). The reconstructed 3D tomograms revealed double-membrane structures that tightly enwrap dense Cvt aggregates and around 74% of them localize as close as 5 nm in three dimensions to other single-bilayer membranes of the cell (Movie EV5). In conclusion, our studies quantitatively describe the three-dimensional shapes and cellular environment of the main stages of Cvt vesicle biogenesis in the cytosol.

## Discussion

By establishing a structural model of the Cvt pathway on multiple levels of resolution, we provide fundamental insights into a prototype example of selective autophagy. Using CLEM, we have unambiguously identified and imaged key stages of Cvt vesicle assembly

in detail: Dense protein aggregates containing Cvt cargo components accumulate in the cytosol and are enclosed by a double membrane that expands and tightly wraps around the cargo (Fig 5D–F and Movies EV2, EV3, EV4 and EV5). The high level of morphological detail in our images and statistical shape measurements exceed previous descriptions at lower resolution [12]. Our data provide direct visual evidence of the proposed concept of exclusive autophagy [24] where tight bending of the membrane around the cargo excludes sequestration of cytoplasmic non-cargo material such as ribosomes. By revealing the close apposition of cargo and phagophore membrane, we captured a crucial scaffolding role of cargo and receptor proteins in mediating efficient and tight binding between the nascent phagophore and cargo.

Our results provide molecular clues to this mechanism as we have found that all main constituents of Cvt vesicles including the receptor Atg19 are organized in defined oligomeric species. Our crystal and EM structures of major cargo mApe1 and secondary cargo Ams1 revealed that they form dodecamers or tetramers, respectively, and thus represent two examples of enzyme complexes that have evolved to self-assemble. The sequestration of enzymes into homo-oligomers represents an assembly state ideally suited for efficient packaging and cargo delivery to the vacuole, while it also ensures exclusivity of the cargo. Su and colleagues revealed that the positioning of the propeptide on a tetrahedral Ape1 cargo is critical for cargo transport via the Cvt pathway by demonstrating that the disruption of oligomeric interfaces prevents the formation of stable dodecamers *in vitro* and the cargo from being incorporated into the Cvt vesicle *in vivo* [31]. Our *in vitro* Atg19 pull-down also suggests a possible role of Ams1 tetramers for selective recognition. Oligomeric structures are essential in other selective autophagy processes. For instance, NCOA4 is a cargo receptor that directly recognizes oligomeric assemblies of ferritin shells in a process that is critical for iron homeostasis [38].

Consistent with the requirement for aggregation and in line with the observations of particle aggregates in our CLEM images, we observed a propensity of major cargo prApe1 dodecamers to form higher-order assemblies in our EM preparations (Fig 1A). In previous experiments, it was not possible to observe the self-aggregating property of prApe1 as it had been solubilized by detergent [31]. In contrast to major cargo prApe1, secondary cargo Ams1 does not exhibit a strong tendency to form higher-order assemblies although EM images of Ams1 reveal occasional chains and larger aggregates alongside single particles (Fig 2A). These modulated biochemical self-interaction properties may be a discriminating feature that defines the role of the secondary cargo Ams1 as opposed to the primary cargo prApe1 in the Cvt assembly. Furthermore, we show that Atg19 regulates the size of the prApe1 higher-order assemblies and that Atg19 interacts with tetramers of secondary cargo Ams1. In this way, the more abundant prApe1 can serve as a vehicle to carry less abundant secondary cargo proteins like Ams1 via Atg19 interaction to the vacuole [39]. Similarly, the selective autophagy receptor p62 forms higher-order structures [40] through self-polymerization [41], which are critical for the targeting of p62 to autophagosomes [42,43]. The additional level of organization beyond soluble oligomers in large molecular assemblies of cargo and receptors appears to be critical to act as transport vehicles to carry additional cargo proteins as well as molecular scaffolds assisting the envelopment of the double membrane.

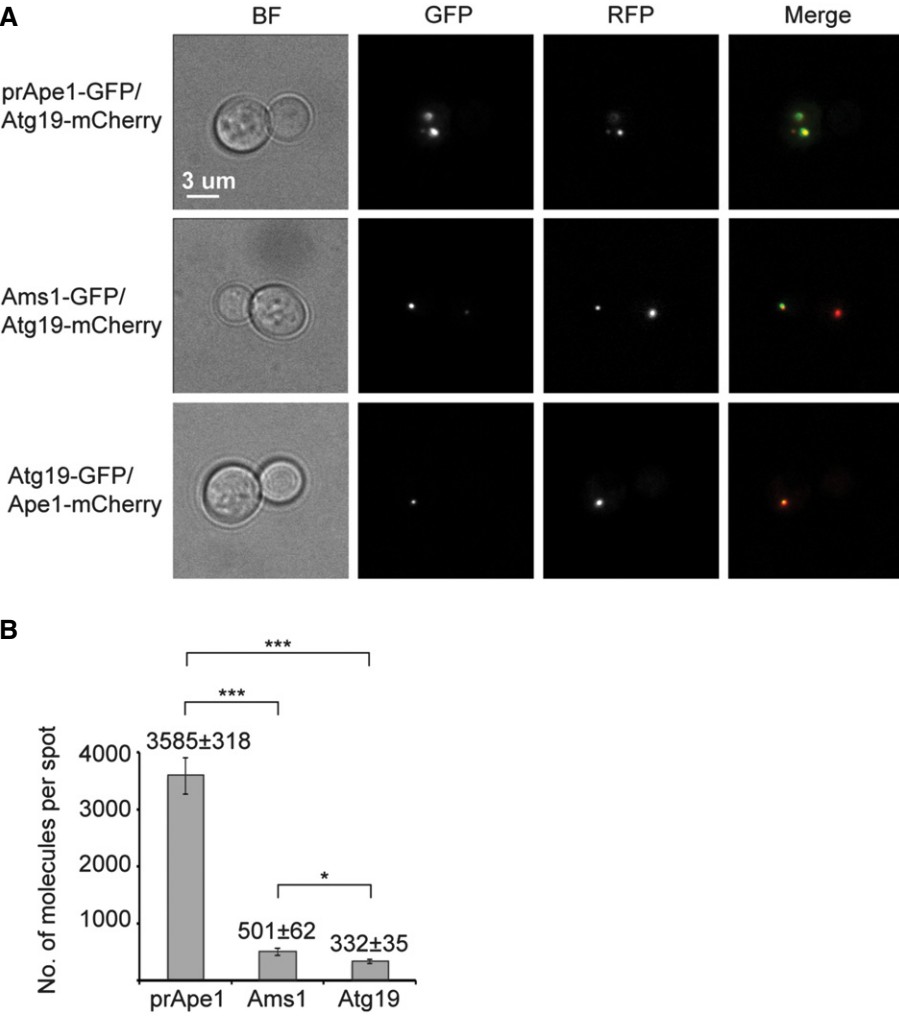

**Figure 4.  Quantification of the Ape1, Ams1, Atg19 protein abundances in living cells.**

A   Epifluorescence microscopy images of the prApe1-GFP/Atg19-mCherry/ypt7Δ, Ams1-GFP/Atg19-mCherry/ypt7Δ, Atg19-GFP/Ape1-mCherry/ypt7Δ *Saccharomyces cerevisiae* cells that were used to quantify the fluorescence intensity of GFP spots. Only colocalizing GFP and RFP spots were considered. BF represents bright-field images.

B   Bar plot showing the number of molecules of prApe1, Ams1, and Atg19 found in Cvt vesicles in living cells. Data are representative of two independent experiments in which a total of 217 prApe1-GFP, 181 Ams1-GFP, and 233 Atg19-GFP spots were analyzed. Error bars indicate standard deviation (SD). ***$P$ < 0.001; *$P$ = 0.0176 (Z test).

Despite the observed tendency of prApe1 to aggregate into large structures, the defined size of Cvt vesicles implies that mechanisms to control cargo assembly exist *in vivo*. Atg19 acts as an autophagy receptor bridging cargo proteins to the autophagy core machinery via Atg11 and Atg8-decorated double membrane of the Cvt vesicle [27]. We have found that Atg19 forms a trimer in solution, raising the question whether the oligomeric platform of Atg19 serves to bind multiple prApe1 dodecamers simultaneously. Our EM observations and co-sedimentation experiments with the prApe1/Atg19 complex indicate that Agt19 interacts with prApe1 dodecamers by competing with propeptide *trans*-interactions as opposed to cross-linking multiple dodecamers. We speculated that this efficient solubilization *in vitro* was the underlying mechanism that controls the size of prApe1 structures *in vivo*. Indeed, our fluorescence microscopy experiments of Atg19

knockout *S. cerevisiae* cells show larger prApe1 puncta, demonstrating a role for Atg19 in regulating the size of prApe1 aggregates and consequently of the Cvt aggregate inside the Cvt vesicle. Interestingly, the ~10-fold excess of prApe1 over Atg19 excludes the possibility of Atg19 saturating all available propeptides on the prApe1 surface. Specifically, when considering the here-determined oligomeric states, it follows that on average one Atg19 trimer is found alongside one Ams1 tetramer and three dodecamers of prApe1 (1:1:3), thus leaving the majority of propeptides available to participate in higher-order prApe1 assemblies. It should be noted that the determined stoichiometric ratio of prApe1, Ams1, and Atg19 *in vivo* is likely to represent an average over a molecule population instead of well-defined complexes. The relatively high ratio of Atg19 suggests an isotropic spatial distribution of the receptor over the Cvt vesicle and disfavors the

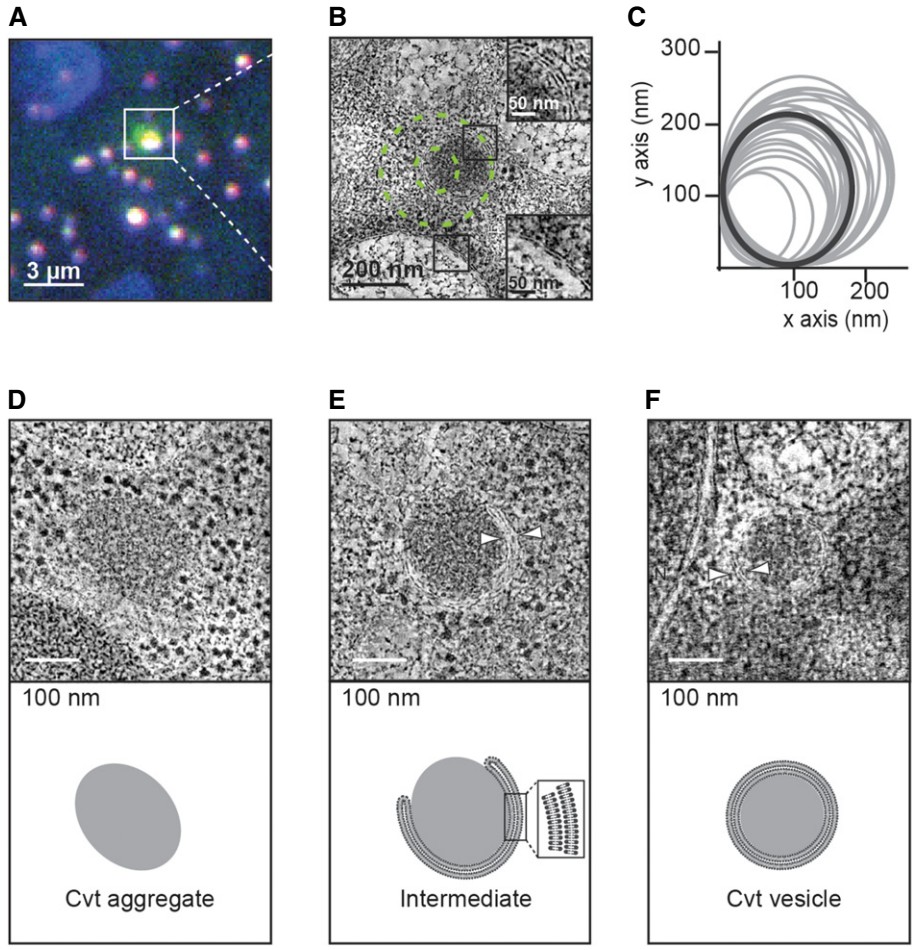

**Figure 5.    Correlative light and electron microscopy of *Saccharomyces cerevisiae* Cvt vesicles from resin-embedded cell sections.**

A    Fluorescence microscopy image of a resin section of prApe1-GFP/Atg19-mCherry/ypt7Δ cells, merge of green (prApe1-GFP) and red (Atg19-mCherry) channels. Tetraspeck beads are detected in the blue channel.

B    Electron tomography slice of the boxed area in (A). The outer and inner dashed green circles of 160 and 60 nm radius represent estimated 50 and 90% localization accuracy, respectively. Insets show close-up views of a double (top) and single (bottom) membrane. Corresponding tomogram is shown in Movie EV2.

C    Plot of the *x* and *y* dimensions of the Cvt vesicles obtained from localization correlation. The average dimensions give rise to spheroids with an average ellipticity of 11% (shown in dark gray). Both prApe1-GFP and prApe1-GFP/Atg19-mCherry datasets were included. The cases representing autophagosomes were excluded.

D    Dense Cvt aggregate that is not surrounded by membrane. Corresponding tomogram is shown in Movie EV3.

E    Cvt aggregate that is partially enwrapped by double membrane. White arrowheads indicate the double membrane. Corresponding tomogram is shown in Movie EV4.

F    Cvt vesicle fully enclosed by double membrane. Bottom: Schematic representation. White arrowheads indicate the double membrane. Corresponding tomogram is shown in Movie EV5.

possibility of Atg19 decorating only the surface of the Cvt aggregate. This interpretation is further supported by previous EM micrographs of immuno-gold-labeled Atg19 receptor [24] and is in line with its capability of solubilizing prApe1 aggregates, thus imposing a mechanism of size control on the Cvt vesicle. Hence, Atg19 is spatially distributed over the entire Cvt vesicle and the relative levels of Atg19 and its cargo are critical to limit the inherent aggregation propensity of prApe1, thereby determining the overall size of the Cvt assembly. In analogy to the recently described mammalian selective autophagy receptor p62 polymers [40], the equilibrium between assembly and disassembly of Cvt aggregates is tightly regulated by its native binding partners.

Together, we have structurally characterized cargo proteins of the Cvt vesicle at various scales of resolution, laid out the

foundations of how the major cargo interacts with its designated autophagy receptor and how the cargos can be placed into the cellular framework of the Cvt pathway. In light of the available literature, our data allow us to redraw the currently prevalent model of Cvt vesicle formation: In the cytosol, prApe1 forms homo-oligomers as dodecamers (Fig 6A) and assembles into higher-order chain-like aggregates via its propeptide (Fig 6B). Atg19 competes with prApe1 propeptide *trans*-interactions, becomes an integral part of the Cvt aggregate, and thus limits aggregate size (Fig 6C). In parallel, Atg19 also recruits Ams1 tetramers to the forming assembly. Hence, Ape1 dodecamer cargos together with Ams1 tetramers and Atg19 trimers assemble into a densely packed core in an approximate stoichiometric ratio of 3:1:1, which determines the size of the forming Cvt

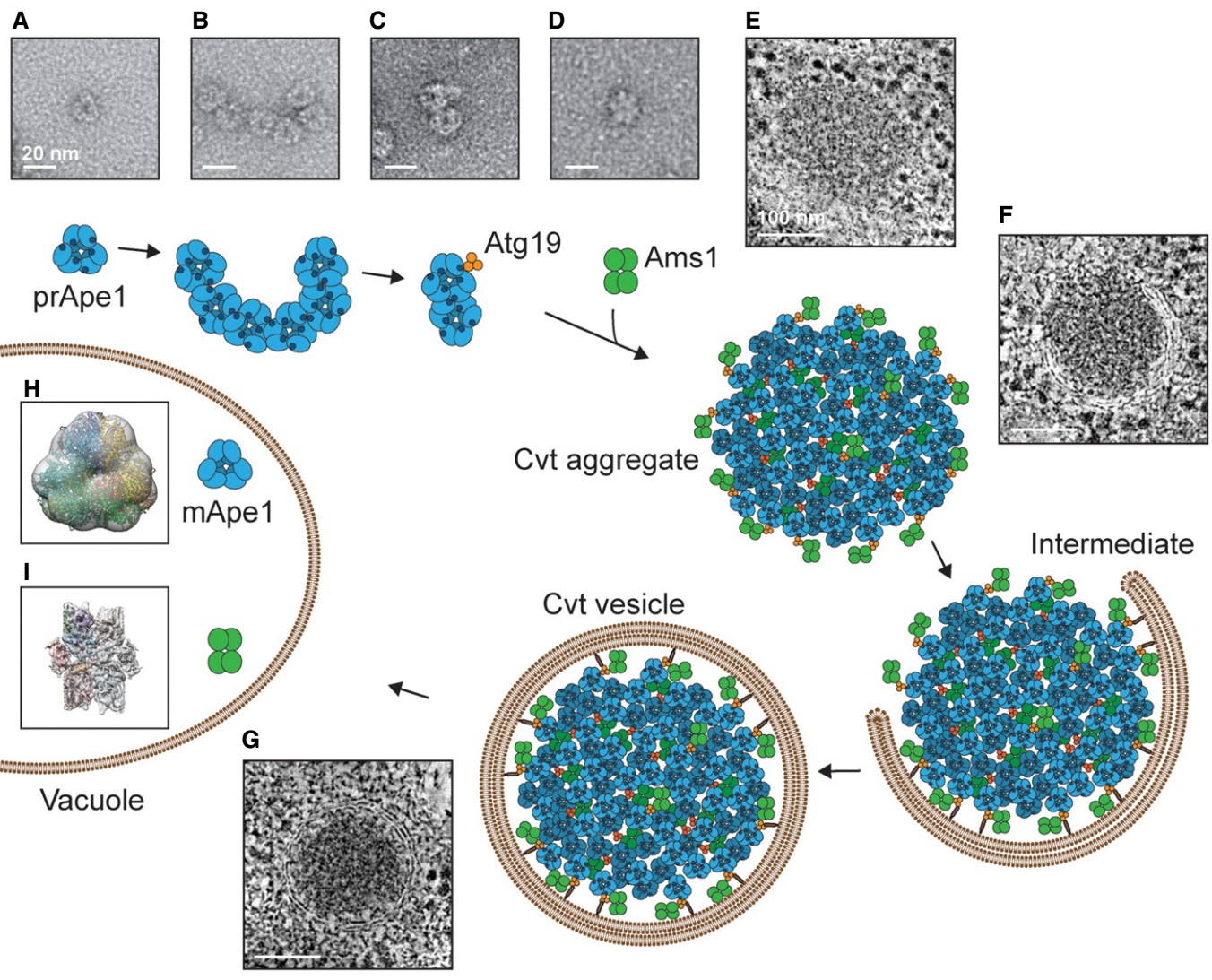

**Figure 6. Model of the Cvt pathway.**

A–I prApe1 (A), the major Cvt cargo, forms homo-oligomers as dodecamers that (B) assemble into higher-order assemblies mediated by the N-terminal propeptide. (C) These assemblies are capable of binding Atg19 that forms a trimer in solution. (D) Ams1, which forms a homo-oligomer as a tetramer, joins the Cvt assembly by binding to Atg19 via a site that is distinct from the Ape1 binding site. (E) prApe1 dodecamers, secondary cargo enzymes such as Ams1 tetramers and Atg19 trimers, are the main constituents of the Cvt aggregate in the cytosol. (F) The Cvt aggregate becomes enwrapped by a double membrane via Atg8 and Atg19 interaction to form (G) the Cvt vesicle. Cvt assembly stages in (E, F, and G) are drawn respecting the determined average size of the vesicles, the dimensions, and stoichiometry of the major cargo prApe1 and secondary cargo Ams1 found in this article. (H) After fusion with the vacuole, prApe1 is subjected to maturation while maintaining the dodecamer organization. (I) Ams1 does not undergo a maturation process in the vacuole.

aggregate (Fig 6E). In conjunction with Atg8/Atg11, the double-membrane phagophore wraps tightly around the Cvt aggregate to enclose cargo and exclude non-cargo material (Fig 6F and G). The ability of cargo and receptor components to self-interact enables efficient packaging and provides an effective transport platform for binding of additional cargo and membrane recruitment through multiple binding sites. The analogy of our example to other selective autophagy pathways from higher eukaryotes [38,40] illustrates that studying the molecular details of cargo assembly and Cvt vesicle formation provides common structural clues to better understand the molecular mechanisms of selective autophagy.

## Materials and Methods

### Cloning, expression constructs, yeast strains, transfection of Sf21 cells

All clones were generated by InFusion cloning, following the manufacturer's protocol (Clontech) unless otherwise stated. *Saccharomyces* (*S.*) *cerevisiae* mApe1, prApe1 and *Chaetomium* (*C.*) *thermophilum* mApe1 (ct-mApe1) cDNAs were cloned into pETM33 to generate N-terminal GST fusion proteins. *S. cerevisiae* Atg19 cDNA was cloned into pETM40 to generate an N-terminal MBP fusion protein. For *Pichia* (*P.*) *pastoris* expression of Ams1, the

pGAPZα vector (Invitrogen) was modified as in [44]. Ams1 mutagenesis was performed using the QuikChange site-directed mutagenesis kit following the manufacturer's protocol (Agilent). To generate vectors for insect cell expression, GST-prApe1, GST-mApe1, and MBP-Atg19 were cloned into pACEBACDual MCS1. Primer sequences are available on request.

For bacterial expression, all constructs were transformed into *E. coli* BL21 RIL cells and grown in lysogeny broth (LB) medium. *P. pastoris* SMD1168 strain (Invitrogen) was transformed with pGAPZ-His$_6$-Ams1 wild-type and W234E mutant as described Watanabe *et al* [44].

*S. cerevisiae* strains for quantitative fluorescence microscopy and CLEM were generated by homologous recombination into the endogenous gene locus with a PCR cassette. C-terminal tagging with EGFP or mCherry as well as deletion strains was generated using pYM28 (GFP), pFA6a-mCherry-KanMX4, and pFA6a-natNT2 cassettes, respectively [45]. Correct integration was verified by DNA sequencing. After insertion of genomic tags, original strains were mated and haploid spores were selected after sporulation. A list of yeast strains generated can be found in Table EV1.

Baculovirus production was carried out following previously described protocols [46]. Briefly, plasmids encoding GST-prApe1, GST-mApe1, or MBP-Atg19 were integrated into baculoviral genome by tn7 transposition via electroporation of DH10EMBacaY cells. Baculoviral DNA was isolated and used to transfect 3 ml of Sf21 cells at a density of $0.3 \times 10^6$ cells/ml using X-tremeGENE HP transfection reagent (Roche). Cells were incubated for 60 h before the supernatant containing baculovirus was removed (V0). V0 virus was then further amplified and used for subsequent large-scale protein expression.

## Protein expression and purification

BL21 RIL cells expressing *S. cerevisiae* or GST-ct-mApe1 were grown at 24°C for 4 h after induction. Cells were lysed by sonication in 1× phosphate-buffered saline (PBS), 500 mM NaCl, 2 mM MgCl$_2$, 1 mM dithiothreitol (DTT), 1 mg/ml lysozyme, and 1× EDTA-free protease inhibitor cocktail (PrInh) (Roche). The supernatant was cleared by centrifugation at 48,000 *g* and supernatant incubated with glutathione beads (GE Healthcare) for 2 h at 4°C before beads were washed extensively and protein eluted by cleavage with GST-3C protease overnight at 4°C. mApe1 dodecamers were separated from other oligomeric states by a continuous 10–30% glycerol gradient run at 146,000 *g* for 16 h at 4°C in a SW 60Ti Beckman rotor. In the case of *S. cerevisiae* mApe1, the GraFix protocol [29] was performed by combining the 10–30% glycerol gradient with a 0–0.15% glutaraldehyde gradient in 50 mM Hepes pH 7.5 and 150 mM NaCl.

Ams1 wild-type and mutant W234E expression in *P. pastoris* was based on described protocols [44]. Expression of *S. cerevisiae* GST-prApe1, GST-mApe1, MBP-Atg19, and the Atg19/prApe1 complex was performed in Sf21 insect cells. First, cells were grown in suspension at 27°C until $1 \times 10^6$ cells and then they were infected or coinfected with virus expressing the protein of interest and harvested 72 h after growth arrest, lysed by sonication, and cleared by centrifugation at 48,000 *g*. In the case of GST-prApe1, lysis was performed in 50 mM Tris–HCl pH 7.5, 25 mM NaCl, 1 mM DTT, 1× PrInh. Lysate was incubated with glutathione beads for 2 h at 4°C, beads washed extensively, and protein eluted in the presence of

50 mM reduced glutathione and 0.1% Triton X-100 before cleavage from GST by incubation with His-3C protease overnight at 4°C. The cleaved protein was further purified by size-exclusion chromatography with a Superose 6, 10/300 GL column (GE Healthcare) run in 50 mM Tris–HCl pH 7.5, 150 mM NaCl, 1 mM DTT. In the case of MBP-Atg19 and the MBP-Atg19/prApe1 co-expression, lysis was performed in 50 mM Tris–HCl pH 7.5, 150 mM NaCl, 5 mM MgCl$_2$, 10% glycerol, 1 mM DTT, and 2× PrInh. The cleared lysate was incubated with amylose beads (New England BioLabs) overnight at 4°C, beads washed extensively before protein was eluted with 10 mM maltose and cleaved with His-TEV protease overnight at 4°C. After cleavage, Atg19 was separated by DEAE ion-exchange chromatography, whereas the Atg19/prApe1 complex was further purified by SEC with a Superose 6, 10/300 GL column run in 50 mM Tris–HCl pH 7.5, 150 mM NaCl, 5 mM MgCl$_2$, 1 mM DTT.

## Biochemical assays

### Activity assay

Ams1 activity was measured according to an established protocol [25,47].

### SEC-MALS

Atg19 was concentrated to approximately 2 mg/ml in 50 mM Tris pH 7.5, 150 mM NaCl, 0.5 mM TCEP. SEC of Atg19 was performed at room temperature on a Superose 6 increase (10/300) column at a flow rate of 0.3 ml/min. Following separation by in-line SEC, the separated sample components were analyzed with a modular triple detector array (Viscotek TDA 305, Malvern Instruments Ltd., Malvern, UK) to determine right-angle light scattering (RALS), refractive index (RI), and UV-vis (UV). The TDA data were processed using Omnisec software. The molecular weight (MW$_{RALS}$) of the species eluting from the SEC column was assessed in triplicates using correlated concentration measurements derived from baseline corrected RI in combination with baseline corrected RALS intensities calibrated against a bovine serum albumin narrow standard (monomeric peak) for both SEC columns.

### In vitro *binding*

Purified MBP-Atg19 (1 μM) or MBP alone (1 μM) was mixed with his-Ams1 (1 μM), mApe1 (1 μM) or prApe1 (0.7 μM—the maximal concentration achievable before protein precipitation) in buffer A (50 mM Tris–HCl, pH 7.5, 25 mM NaCl, 5 mM MgCl$_2$, 1 mM DTT, and 0.2% NP-40) and incubated overnight at 4°C with gentle agitation. Equilibrated amylose beads (125 μl) were added to each reaction and incubated for 2 h at 4°C with gentle agitation. The amylose beads were sedimented by centrifugation and washed four times with 1 ml buffer A (3,500 *g*, 2 min at 4°C). Bound proteins were eluted from the beads by boiling in SDS loading buffer, and the eluates were analyzed by SDS–PAGE followed by staining with Coomassie Brilliant Blue.

### Co-sedimentation

Equivalent amounts of purified *S. cerevisiae* prApe1, prApe1/Atg19, and Atg19 as control were centrifuged in a Beckman TLA100 rotor for 3 h at 186,000 *g* at 4°C. Equivalent volumes of pellet and supernatant were analyzed by SDS–PAGE followed by staining with Coomassie Brilliant Blue.

### Western blot

prApe1-GFP/Atg19-mCherry/Δypt7 and prApe1-GFP/Atg19Δ/ypt7Δ *S. cerevisiae* cells were grown at 30°C until $OD_{600 \, nm} = 1–1.2$. Cells were washed twice in ice-cold PBS 1× and immediately resuspended in 50 mM Tris–HCl pH 8, 100 mM NaCl, 2.5 mM $MgCl_2$, 0.25% Triton X-100, 1 mM PMSF, 1 mM DTT, 1× PrInh. Cells were subsequently lysed with glass beads using FastPrep®-24 (MP BIOMEDI-CALS) set with following parameters: four cycles at speed 4, 15 s each with 3 min cooling in between. Lysates were spun at 17,000 *g*, 4°C to remove cell debris. Equivalent amounts of clarified lysates were subjected to western blot using anti-Atg19 antibody (polyclonal, 1: 200), anti-Ape1 antibody (polyclonal, 1:1,000), and anti-tubulin antibody (TAT1, Abcam, monoclonal, 1:500).

### Negative stain image reconstruction

Negatively stained specimens (sc-mApe1 and Ams1) for electron microscopy (EM) were prepared by the droplet technique with 2% uranyl acetate and imaged using a Philips CM-120 transmission electron microscope, operated at 120 kV, and equipped with a TVIPS 4k × 4k CCD camera. For GraFix-purified sc-mApe1, 30 micrographs were collected at an underfocus between 1.0 and 1.5 μm with a nominal magnification of 53,000 corresponding to 1.9 Å pixel size. A total of 5,481 particles were manually picked using E2BOXER of the EMAN2 package [48] and class averages were generated using IMAGIC and SPIDER [49,50]. We built the initial models, first by aligning a set of 20 best-defined classes against a Gaussian blob while imposing tetrahedral symmetry, second by assigning the corresponding Euler angles manually and imposing tetrahedral symmetry. Both initial models converged to the same 24 Å resolution structure after 20 cycles of iterative refinement with SPIDER. For an initial 3D model of Ams1, a random-conical tilt dataset was acquired on an FEI Polara microscope operated at 100 kV at an underfocus of 1.8 μm with a magnification of 59,000 corresponding to 1.91 Å pixel size on a US4000 Gatan CCD camera. A total of 1,830 tilt-pair particles were selected, subjected to 2D classification, and used for a reconstruction using the EMAN2 random-conical tilt protocol [48] while imposing D2 symmetry. Subsequently, we further refined the structure using 9,865 untilted particles in the SPIDER software suite to a resolution of approximately 31 Å.

### Crystallization, model refinement, and homology modeling

Small bipyramidal crystals (5 × 5 × 5 μm) of ct-mApe1 (2.0–3.0 mg/ml in 50 mM Tris–HCl, 50 mM NaCl, pH 7.5) were grown using hanging drop vapor diffusion at 4°C after 3–7 days in 100 mM Hepes (pH 6.6), 4 M sodium formate. For cryoprotection, crystals were soaked for 5 min in 100 mM Tris–HCl pH 8.5, 4.6 M sodium formate, 2 mM $ZnCl_2$ supplemented with 15% (v/v) glycerol and flash-cooled in liquid nitrogen. Diffraction data were collected at the ID23-2 and EMBL P14 microfocus beamlines at European Synchrotron Radiation Facility and at the DESY PETRA III storage ring, respectively, and processed with XDS [51] and SCALA [52]. Self-rotation functions revealed four non-crystallographic threefold axes and eight non-crystallographic twofold axes, showing the asymmetric unit is composed of six ct-mApe1 subunits and dodecamers are generated by crystallographic symmetry. The crystal structure was solved by molecular replacement in Phaser [53] using a mammalian tetrahedral aspartyl aminopeptidase model (PDB 3vat) with all side chains truncated to $C_\beta$ atoms. An anomalous difference Fourier electron density map calculated using data from the K absorption edge for zinc revealed strong density at both metal sites. The molecular replacement solution was refined through iterative rounds of reciprocal-space refinement in Phenix [54] and manual rebuilding in Coot [55]. Positional non-crystallographic symmetry constraints were imposed throughout the refinement and zinc-ligand distances were restrained according to Harding [56,57]. Table 1 summarizes data collection and refinement statistics.

### Single-particle electron cryomicroscopy and atomic model building

To optimize the dispersity of the sample, $His_6$-Ams1 at 0.4 mg/ml was dialyzed into a 50 mM Tris–HCl, 175 mM NaCl, 75 mM imidazole buffer. The sample was applied to glow-discharged 300 mesh Quantifoil R 2/2 grids, plunge-frozen in liquid ethane using an FEI Vitrobot, and transferred to an FEI Titan Krios microscope operating at 300 kV. Micrographs were recorded using EPU and a Falcon II direct electron detector at an underfocus between 1 and 5 μm with a total dose of 58 $e^-/Å^2$, accumulated in seven frames at a final pixel size of 1.084 Å. For preprocessing, we used MotionCorr [58] and determined the contrast transfer function parameters using CTFFIND3 [59]. Subsequent processing steps of 3D structure refinement of the data set were conducted with RELION-1.3 [60]: 85,274 particles were subjected to 2D classification followed by 3D structure refinement, and a homogeneous subpopulation of 33,588 particles was selected based on 3D classification and further processed using the particle polishing procedure, which resulted in a final 6.3 Å resolution map based on the 0.143 Fourier shell correlation criterion. The obtained map was sharpened by applying a B-factor of $-80 \, Å^2$ and filtered to 5.0 Å.

The complete quasi-atomic model of Ams1 was built by combining homology and *de novo* domain models followed by flexible EM density-guided fitting. First, a homology model of the well-conserved C-terminal portion (287–1,083) was computed using MODELLER [61] based on the structure of *Streptococcus pyogenes* α-mannosidase (PDB 2wyh) with a sequence identity of 17%. Second, due to the lack of closely related structural templates for the N-terminal part (1–286), Ams1 was subjected to a search for modeling templates using MODexplorer (http://modorama.org, [62] and Genesilico Metaserver (https://genesilico.pl/meta2, [63]). As a result, for residues 45–203, we identified a jelly-roll fold from RetS periplasmic sensor domain (PDB 2xbz, chain A) with significant similarity scores to Ams1 (HHSearch [64] probability up to 96%) compatible with the EM density. We could further improve the visual match between the structure and EM density by introducing fragments 45–55 and 114–127 from a putative β-galactosidase from *B. fragilis* (PDB 3 cmg, chain A). However, for the remaining N-terminal residues 1–44 and residues 204–286, no modeling template could be identified while the unassigned part of the cryo-EM map revealed four tubular densities forming an apparent four α-helix bundle. In support, α-helical structure predictions of residues 209–271, they were modeled *de novo* using Rosetta AbinitioRelax [65], by generating 1,000 alternative models and selecting the model of highest cross-correlation with the EM map. Finally, we assigned the

remaining helical density to a predicted N-terminal helix for residues 17–27, and due to uncertainty of the sequence register, this helix was built as an ideal poly-alanine helix. To resolve steric clashes and geometry deviations, the combined structures were energy-minimized with GROMACS [66] prior to flexible fitting with the DireX software [67]. The C-terminal (residues 287–1,083) and N-terminal regions (17–286) were fitted separately against map fragments carved from the sharpened EM map. It should be noted that the N-terminal four-helix bundle could alternatively be connected to the α/β barrel of the adjacent subunit (Fig 2K, chain D). The distance of the connecting residues and compactness of fold, however, favored the connectivity of our current model. To limit overfitting, we applied secondary structure restraints from the reference structures. We evaluated a grid parameter search (den_strength, den_gamma, pert_fac, den_strength_loop) by identifying solutions with highest map correlation along with physically plausible stereochemistry and clash scores. Finally, for our selected model, we applied the parameter values 0.01, 0.8, and 0.06 for pert_fact, den_strength, and den_gamma. To decrease the effect of the reference structure for regions outside predicted secondary structure elements, we scaled the den_strength for loop regions by a factor of 0.35. With the application of D2 symmetry, we then generated the remaining monomers and the tetramer was refined against the full map with strong restraints on the monomer structure. A final energy minimization step with restraints on the main chain atoms was performed to resolve deviations from reference stereochemistry arising from the DireX refinement.

**Quantitative fluorescence microscopy**

Cells were grown in SC-Trp medium at 30°C till $OD_{600 \text{ nm}} = 0.6$. prApe1-GFP, Ams1-GFP, and Atg19-GFP cells were individually mixed 1:1 with Nuf2-GFP cells of the same mating type. Cells were incubated for 10 min at room temperature to adhere on concanavalin A-coated coverslips and then washed with SC-Trp medium. Cells were imaged in 40 µl of SC-Trp medium at room temperature with an Olympus IX81 wide-field epifluorescence microscope, equipped with at 100×/1.45 objective and a Hamamatsu Orca-ER CCD camera. Samples were excited with a X-Cite 120Q lamp (Olympus) at 100% of power for 100 ms for the GFP channel and 250 ms for the RFP channel. For each channel, the samples were imaged as z-stacks of 23 frames (Fig 3H) or 21 frames (Fig 4A) spaced by 200 nm each. The stacks were acquired one frame at the time for both channels. All the microscope setup was controlled through Metamorph 7.5 (Molecular Devices). For Fig 3H, GFP patches of prApe1-GFP/Atg19Δ/ypt7Δ cells were quantified as follows: The background in the cells was subtracted by median filtering (kernel = 20 pixels). The intensities of the patches were measured from a rectangular selection in the frame of the z-stack where the patch was brightest. The size of the rectangular selection was large enough to surround the thresholded patch and was allowed to vary, to accommodate the heterogeneity of patch sizes. The median intensity of prApe1-GFP was normalized to the median intensity of Nuf2-GFP spots. prApe1-GFP patches in prApe1-GFP/Atg19-mCherry/ypt7Δ cells were quantified for comparison and were subjected to identical procedure for the quantification and normalization of the spot intensities. The error associated with each median was the standard error for the median (SEM) calculated as:

$$\sigma = \frac{1.4826 \exp{(I)} \, \text{MAD}}{\sqrt{N}}$$

where MAD is the median absolute deviation computed on the log transformation of the measured fluorescence intensities ($I$). The error of the intensities normalized to the number of Nuf2 molecules was computed propagating the standard errors for the medians accordingly. For Fig 4B, the quantification of the patches was done by a custom-written software in Python 2.7 as detailed in [36]. In brief, the number of prApe1-GFP, Ams1-GFP, or Atg19-GFP molecules was quantified using Nuf2 molecules as a reference. For each kinetochore patch, 280.6 ± 16.1 molecules of Nuf2 were counted. The number of Nuf2 molecules was quantified by comparing Nuf2 and Cse4 and by considering five molecules of Cse4 for kinetochore (Lawrimore *et al* [35], Picco *et al* [36]). The image stacks were background-corrected by subtraction of the median-filtered image to the image itself. The fluorescence intensity of the patches was measured by integrating the fluorescence intensity of each patch through the frames of the stack. Patches at the beginning or at the end of the stack were discarded. For Ams1, Ape1, and Atg19, only the patches that were colocalizing with an mCherry marker were quantified: Atg19-mCherry for Ams1 or Ape1 and Ape1-mCherry for Atg19.

**Correlative light and electron microscopy**

Sample preparation and data collection for the prApe1-GFP/ypt7Δ dataset were performed as described previously [37,68,69], with minor modifications for the prApe1-GFP/Atg19-mCherry/ypt7Δ dataset. Briefly, cells were grown in YPAD medium at 30°C to an $OD_{600} = 0.6$, high-pressure-frozen, freeze-substituted, embedded in Lowicryl HM20, and sectioned to 300 nm. Grids were incubated with 50 nm TetraSpeck™ beads (Life Technologies) and imaged with an Olympus IX81 wide-field epifluorescence microscope equipped with a 100×/1.45 objective and a Hamamatsu Orca-ER CCD camera. Grids were subsequently incubated with protein A-coated gold beads, stained with Reynolds lead citrate, and electron tomography was performed on an FEI F30 TEM operated at 300 KV, equipped with an FEI Eagle 4K CCD camera and a dual tilt holder (Fischione Model 2040), using the SerialEM software [70]. Tomograms were reconstructed using the IMOD 4.7.13 software package [71]. The position of the fluorescent structures of interest was determined using the correlation procedure described before [37,69].

**Accession numbers**

The electron microscopy maps of *S. cerevisiae* mApe1 and Ams1 have been deposited in the EM Data Bank and available with accession codes (EMD-8167, EMD-8166). Atomic coordinates of *C. thermophilum* mApe1 and *S. cerevisiae* Ams1 have been deposited at the Protein Data Bank under accession codes (5JM6, 5JM0).

**Expanded View** for this article is available online.

## Acknowledgements

We thank the EMBL Heidelberg Protein Expression and Purification Core Facility for maintaining the insect cell culture facility. We acknowledge technical support by Jaclyn Chan. We also appreciate the support of the EMBL EM facility.

We are grateful to IT Support from F. Thommen and M. Wahlers for setup and maintenance of the high-performance computational environment. In addition, we express gratitude to Michael Meurer and Michael Knop for yeast plasmids, Ambroise Desfosses for help with EM image processing, and Alejandro Reyes for advice on statistical evaluation. A.J.J. acknowledges support by Marie-Sklodowska-Curie (MSC) Fellowship (PIEF-GA-2012-331285). C.B. and Y.S.B. acknowledge support by the EMBL International PhD Program, S.S. by the MSC ITN RNPnet (PITN-GA-2011-289007), A.J.J. and J.K. by postdoctoral fellowships from the EMBL Interdisciplinary Postdoc Program (EIPOD) under MSC COFUND actions (PCOFUND-GA-2008-229597), and W.K. by a fellowship of the Swiss National Science Foundation.

## Author contributions

CB, SS, and CS designed the project. CB, AJJ, MW, and CS determined the mApe1 crystal and low-resolution EM structure. SS, WJHH, AJJ, JK, and CS determined the Ams1 cryo-EM structure. CB, AP, and MK measured the *in vivo* stoichiometry by quantitative fluorescence microscopy. CB, YSB, WK, and JAGB performed CLEM experiments. CB, SS, ACR, AJJ, and AKT carried out biochemical experiments. CB, SS, AKT, AJJ, and CS wrote the manuscript with support from all authors.

## Conflict of interest

The authors declare that they have no conflict of interest.

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
