## [Review Process File · EMBO Reports]

Manuscript EMBO-2015-41960

Higher-order assemblies of oligomeric cargo receptor complexes form the scaffold of the Cvt vesicle

Chiara Bertipaglia, Sarah Schneider, Arjen Jakobi, Abul Tarafder, Yury Bykov, Andrea Picco, Wanda Kukulski, Jan Kosinski, Arvind Ravichandran, Wim Hagen, Matthias Wilmanns, Marko Kaksonen, John Briggs, Carsten Sachse

Corresponding author: Carsten Sachse, EMBL - European Molecular Biology Laboratory

Review timeline:

Decision from The EMBO Journal:	09 December 2015
Transfer to EMBO reports:	23 December 2015
Editorial Decision:	06 January 2016
Revision received:	06 April 2016
Editorial Decision:	20 May 2016
Revision received:	29 April 2016
Editorial Decision:	02 May 2016
Accepted:	02 May 2016

Editors: Achim Breiling/Martina Rembold

Transaction Report:

Decision from The EMBO Journal

09 December 2015

Thank you for submitting your manuscript to us. I have now received reports from two good referees, which I enclose below. I am afraid that their recommendation let us to conclude that we cannot offer publication in The EMBO Journal.

As you will see, the referees appreciate the quality of your work and realize that it adds to the previously published study and will as such be of interest to the field of selective autophagy. However, while referee #2 offers some support, referee #1 thinks that in light of the recent Su et al. paper, the overall advance at this stage remains too limited for further consideration here. Referee #2 furthermore notes that the models you derive from the structural analyses are not functionally tested. Given that neither referee strongly endorses publication in The EMBO Journal, I am very sorry to say that I have no other choice but to return your manuscript to you.

I thank you in any case for the opportunity to consider this manuscript. I am really very sorry I cannot be more positive this time. I hope that our referees' comments are nevertheless helpful, and that you will find a good home for your work.

REFeree REPORTS

Referee #1:

This is a well-executed study of the molecular and cellular structure of the yeast Cvt (selective autophagy) substrates Ape1 and Ams1. Unfortunately for the authors, most of the main results were already published in the journal *Autophagy* by Su et al. & Chang (on line in July, in print Sept. 2015). The crystal and EM structures of Ape1 and the evidence that the propeptide drives aggregation were included in the Su et al. paper. The main conclusions and model shown in Fig. 6 of this paper were already evident from the Su et al. study, if not as elegantly presented. Bertipaglia et al. add to this with the cryoEM structure of Ams1, copy number quantitation of prApe1, Ams1, and Atg19, and CLEM of Cvt vesicles. The Ams1 structure is the most significant piece of new data. The cellular microscopy work is beautiful but it is not clear what it adds conceptually to the field.

Major

The Ams1 structure is the main new finding, however the findings are not integrated into the functional studies. A model is presented in Fig. 6 for the recruitment of Ams1 into prApe1 aggregates, but the mechanism of Ams1 recruitment is not actually explored experimentally. If this were pursued, it would make a stronger case that this study contains a significant advance over what was already published by Su et al.

The sections that re-state results of Su et al. could be eliminated or shortened.

Specific points

The authors need to be more clear about what they are using as a reference for quantitation of molecular copy number. Presumably they used Nuf2, but this itself is calibrated using Cse4, the quantification of which shows some variability from paper to paper.

The validation of the EM reconstructions and the documentation provided for review are not up to current standards. Tilt pair analysis, particle orientation distributions, and model correlation coefficients need to be shown. The maps and models need to be provided for review.

Referee #2:

The manuscript by Bertipaglia et al. reports the structural characterization of autophagic cargo - receptor complexes in order to obtain insights into the process of selective autophagy. To this end the authors have focused on the *S. cerevisiae* cytoplasm-to-vacuole-transport (Cvt) pathway during which the oligomeric prApe1 protease as well as the glycosidase Ams1 are transported into the vacuolar lumen. This transport requires the Atg19 cargo receptor, which binds the cargo proteins and links them to the membrane of the Cvt vesicles. prApe1 is initially synthesized with a N-terminal pro-peptide that is cleaved off in the lumen of the vacuole to generate the active mApe1 protease. Although the Cvt pathway is a specialized pathway in *S. cerevisiae* it has served as model pathway to uncover factors and principles underlying selective autophagy in general. In fact many genes required for autophagy also in higher eukaryotes were identified using the Cvt pathway as model. Furthermore, the concept of cargo receptors was first introduced by analysis of this pathway.

Bertipaglia et al. determined the structure of the *S. cerevisiae* prApe1 particle using negative stain EM and also determine the crystal structure of mApe1 from *C. thermophilum* using x-ray crystallography. The crystal structure of the *S. cerevisiae* mApe1 was recently published by another group (Su, 2015, *Autophagy*). The structure of the Ams1 cargo determined by cryo-EM is also presented. The authors then show that the Atg19 cargo receptor directly binds to prApe1 particles (it was already known that it binds to the prApe1 pro-peptide (Scott et al., 2011, *Mol Cell*; Sawa-Makarska et al., 2014, *NCB*)). Interestingly the presence of Atg19 appears to restrict the size of the prApe1 particles suggesting that Atg19 might limit the size of the Cvt particles in vivo. Employing quantitative fluorescence microscopy the authors went on to determine the number of prApe1, Ams1 and Atg19 molecules in Cvt vesicles. Finally, the authors use correlative light and electron microscopy to reveal intermediates during Cvt vesicle formation.

In summary the manuscript represents an interesting and thorough structural investigation of Cvt particle/vesicle formation. It will be of interest for scientists working in the field of selective autophagy or membrane dynamics.

Specific comments

1. One of the shortcomings of the manuscript is that the models derived from the structural data were not tested *in vivo*. Most of the structural data may not be of high enough resolution to introduce targeted mutations. However, the authors propose that Atg19 limits the size of the prApe1 particles. Thus, does overexpression of Atg19 result in smaller Cvt particles? This could be quantified using the method employed for Figure 4.
2. In Figure 3D a SEC run of the prApe1 - Atg19 complex is shown. The profiles of the individual proteins should also be shown. Also, could the authors use this method to show that the prApe1 particles become smaller in the presence of Atg19?
3. Figure 3E-F show the negative stain EM analysis of prApe1 - Atg19 complexes. The quantification strongly suggests that Atg19 is preferentially associated with smaller (i.e. single prApe1 particles). Surprisingly, although the prApe1 - Atg19 is very stable (Figures 3C, D) only 20% of the prApe1 particles show density for Atg19. Where does this discrepancy derive from?
4. In Figure 3B the authors show the bead fraction of a MBP pull down assay. The input fraction should also be shown.
5. The authors write in the introduction "Currently, more than 35 proteins have been shown to be involved in this process (Mizushima et al., 2011)". However, to my knowledge there are currently 41 Atg proteins (Yao et al., 2015, Autophagy).
6. On page 10 the authors refer to Figure S5 instead of Figure S4.

TRANSFER TO EMBO REPORTS

23 December 2015

1st Editorial Decision

06 January 2016

Thank you for the transfer of your research manuscript to EMBO reports. I apologize for the delay in getting back to you, which is due to the Christmas holidays during which our editorial offices were closed.

I appreciate that your study extends previous knowledge on the Cvt pathway by analyzing the pathway with two cargos, Ape1 and Ams1. Based on the comments of referee 1 and 2, who reviewed the manuscript for the EMBO Journal, and your point-by-point response I would like to give you the opportunity to submit a modified and strengthened version of your work.

From the analysis of the referee comments it becomes clear that further functional data has to be added to support the model and proposed pathway derived from the structural data. In particular, the proposed structure-function analysis to explore the mechanism of Ams1 recruitment should be provided to address point 4 of referee 1 and point 1a of referee 2. Moreover, it has to be investigated if Atg19 restricts the size of prApe1 particles (point 1b of referee 2). All control experiments have to be provided (point 2a and point 4 of referee 2). The X-ray and EM structure of Ape1 should remain part of the manuscript (referee 1, point 5), but might be shortened if possible.

Please address all referee concerns in a complete point-by-point response. Acceptance of the manuscript will depend on a positive outcome of a second round of review. It is EMBO reports policy to allow a single round of revision only and acceptance or rejection of the manuscript will therefore depend on the completeness of your responses included in the next, final version of the manuscript.

Revised manuscripts should be submitted within three months of a request for revision; they will otherwise be treated as new submissions. Please contact us if a 3-months time frame is not sufficient

for the revisions so that we can discuss the revisions further. Please note that all materials and methods should be included in the main manuscript file.

Supplementary/additional data: The Expanded View format, which will be displayed in the main HTML of the paper in a collapsible format, has replaced the Supplementary information. You can submit up to 5 images as Expanded View. Please follow the nomenclature Figure EV1, Figure EV2 etc. The figure legend for these should be included in the main manuscript document file in a section called Expanded View Figure Legends after the main Figure Legends section. Additional Supplementary material should be supplied as a single pdf labeled Appendix. The Appendix includes a table of content on the first page, all figures and their legends. Please follow the nomenclature Appendix Figure Sx throughout the text and also label the figures according to this nomenclature. For more details please refer to our guide to authors.

Regarding data quantification, can you please specify the number "n" for how many experiments were performed, the bars and error bars (e.g. SEM, SD) and the test used to calculate p-values in the respective figure legends? This information is currently incomplete and must be provided in the figure legends. Please also include scale bars in all microscopy images.

We now strongly encourage the publication of original source data with the aim of making primary data more accessible and transparent to the reader. The source data will be published in a separate source data file online along with the accepted manuscript and will be linked to the relevant figure. If you would like to use this opportunity, please submit the source data (for example scans of entire gels or blots, data points of graphs in an excel sheet, additional images, etc.) of your key experiments together with the revised manuscript. Please include size markers for scans of entire gels, label the scans with figure and panel number, and send one PDF file per figure or per figure panel.

I look forward to seeing a revised version of your manuscript when it is ready. Please let me know if you have questions or comments regarding the revision.

1st Revision - authors' response

06 April 2016

We thank the Reviewers for their comments and have provided a point-by-point reply to the questions raised. Here, we present how we addressed the comments of Reviewer #1 and #2 point by point for an *EMBO Reports* article. We have significantly extended the Results, Figures and Discussion of the manuscript based on the requests of the Reviewers.

Referee #1:

This is a well-executed study of the molecular and cellular structure of the yeast Cvt (selective autophagy) substrates Ape1 and Ams1. Unfortunately for the authors, most of the main results were already published in the journal *Autophagy* by Su et al. & Chang (on line in July, in print Sept. 2015).

1. The crystal and EM structures of Ape1 and the evidence that the propeptide drives aggregation were included in the Su et al. paper. The main conclusions and model shown in Fig. 6 of this paper were already evident from the Su et al. study, if not as elegantly presented.

We respectfully disagree with the assessment of Reviewer #1. First, we believe our manuscript has a far greater scope than the Su et al. paper, which focused primarily on the Ape1 cargo whereas our paper, using a greater breadth of methodology, examines the Cvt pathway as a whole on two cargos including their receptor at various levels of resolution. Second, the main conclusions of our article as they are summarized in Figure 6 go substantially beyond the results presented in Su et al. In order to demonstrate this we compared the main conclusions of the papers in Table 1 in a side-by-side manner.

Table 1: Side-by-side comparison of main results from the Su et al. paper, 2015 and our submitted Bertipaglia et al. paper. Overlapping conclusions are highlighted in bold.

Su et al. 2015 Autophagy	Bertipaglia et al. 2015
X-ray crystal structure of S. cerevisiae mApe1 (2.5 Å resolution); Negative stain EM structure of detergent solubilized S. cerevisiae prApe1	X-ray crystal structure of C. thermophilum mApe1 (2.7 Å resolution); Negative stain EM structure of S. cerevisiae mApe1
Isolated propeptide forms tetramer	Negative staining micrographs of chain-like prApe1 assemblies
Structure-based insertion mutant disrupts Ape1 dodecamer in vivo and in vitro	-
Tetrahedral organization of cargo and propeptide positioning required for delivery to vacuole	-
-	Cryo-EM structure of tetramer Ams1 (6.3 Å resolution) including structure-based mutational analysis for interface validation
-	Atg19 forms a trimer in solution
-	prApe1-Atg19 complex formation by pull-down, size-exclusion chromatography, co-sedimentation assay and negative staining EM showing competition between prApe1 assembly formation and prApe1-Atg19 binding, Atg19 knockout cells show larger prApe1 puncta suggesting that Atg19 exerts size control over the Cvt aggregate
-	In vivo stoichiometry of prApe1, Ams1 and Atg19 molecules in Cvt vesicles
-	Morphological analysis of Cvt vesicles in yeast including unprecedented quantification from correlative light and electron microscopy

It is clear from this comparison that there is some overlap between the two papers in terms of the structure of Ape1, but overall limited overlap with our entire manuscript. In numbers, this may represent less than a fifth of the results presented in our manuscript. Given the previous publication in August 2015 of the Su et al. paper, we still feel it is important to validate the presented crystal structure and provide structural differences not covered in their data.

Comparison of the data regarding the Ape1 structure in closer detail reveals even more differences in the conclusions drawn from the Ape1 structures, which we raise in Table 2 below.

Table 2: Detailed comparison of Su et al. 2015 and submitted Bertipaglia et al. paper with respect to the structure and aggregation properties of Ape1.

Su et al. 2015 Autophagy	Bertipaglia et al. 2015
S. cerevisiae mApe1 structure with active site partially disordered due the absence of Zn ²⁺ ions. Determination of a putatively inactive conformation.	C. thermophilum mApe1 structure with two Zn ²⁺ ions coordinated by the active site including a selectivity loop in the adjacent ancillary domain protruding into the active site. Determination of a potentially catalytically active conformation.
Bulk-aggregation of prApe1 after cleavage of solubility tag. Only detergent-solubilized prApe1 as single particles purified and imaged by negative staining EM. No evidence presented of higher-order assemblies.	Full-length native prApe1 purified and imaged as chain-like assemblies of dodecamers in direct comparison to mApe1 dodecamers. Propeptide is characterized in the context of the full-length protein and responsible for the formation of higher-order assemblies.

2. Bertipaglia et al. add to this with the cryoEM structure of Ams1, copy number quantitation of prApe1, Ams1, and Atg19, and CLEM of Cvt vesicles. The Ams1 structure is the most significant piece of new data.

We thank Reviewer #1 for the assessment of novelty for the Ams1 structure. Nevertheless, it is clear from point 1 above that a multitude of other data is included in the manuscript, which led us to propose a refined model of the Cvt pathway.

3. The cellular microscopy work is beautiful but it is not clear what it adds conceptually to the field.

The original cellular electron microscopy work on Cvt vesicles dates back to 1997 Baba et al. from the Ohsumi lab. In our article, we go significantly beyond the previous EM data. We present for the first time an in-depth and quantitative three-dimensional analysis of key stages of the Cvt pathway. While the results do match the current scientific view of the Cvt pathway, this is the first time that they are shown at such a level of resolution in space and time, i.e. 3D tomographic data on a large population of Cvt vesicles (38 vesicles). The data provide strong visual evidence for the mechanism of cargo encapsulation and delivery by the autophagy-related Cvt pathway.

Major

4. The Ams1 structure is the main new finding, however the findings are not integrated into the functional studies. A model is presented in Fig. 6 for the recruitment of Ams1 into prApe1 aggregates, but the mechanism of Ams1 recruitment is not actually explored experimentally. If this were pursued, it would make a stronger case that this study contains a significant advance over what was already published by Su et al.

We agree with Reviewer #1 that additional biochemical data is beneficial to validate the structure and better understand the role of secondary cargo Ams1 in the context of the Cvt complex.

Therefore, we introduced structure-based mutants to disrupt the tetrameric interface of Ams1. Interestingly, we found that a W234E mutant leads to the formation of dimers instead of tetramers while retaining enzymatic activity. Figure 2 has been complemented by micrographs and class averages of the mutant (Figure 2L) and new activity assays and gel filtration chromatograms have been included (Figure EV3C and EV3D respectively). Furthermore, we show that wildtype Ams1 tetramers are able to bind the Atg19 receptor whereas the W234E dimers lose this ability (Figure EV3E). The absence of structural information on the prApe1-Ams1-Atg19 assembly currently precludes more detailed insight into the cargo assembly mechanism. While we feel this exceeds the scope of this study, it remains of high interest for future experiments. The following paragraph has been added to page 7 of the revised manuscript:

“To validate whether the surface formed by the four-helix bundle is responsible for mediating longitudinal contacts within the tetramer, we generated a W234E mutant by introducing a negatively charged residue to selectively disrupt the oligomeric interface (Figure 2K). While the Ams1 W234E mutant is still catalytically active (Figures EV3C), size-exclusion chromatograms show a peak shift towards smaller molecular species in comparison with wildtype Ams1 (Figure EV3D). When the fractions were analyzed by negative staining EM, we predominantly observed smaller particles rather than wildtype tetramers (Figure 2L). Classification of the W234E particles revealed smaller two-winged oligomers of 13 x 10 nm dimensions corresponding to approximately half the size of tetrameric Ams1. Comparison with simulated reprojections of putative Ams1 dimers strongly suggests a dimeric Ams1 complex with longitudinal contacts disrupted and transverse contacts still maintained (Figure 2L). Moreover, in order to test whether these Ams1 samples have the capability of binding the native receptor Atg19, we subjected Ams1 wildtype tetramers and W234E dimers to an Atg19 pull-down assay. The corresponding SDS-PAGE gel shows that wildtype tetrameric Ams1 binds MBP-Atg19 efficiently whereas the introduction of a W234E mutation drastically reduces binding (Figure EV3E). This suggests that the Ams1 wildtype tetramer is required for *in vitro* recognition by the autophagy receptor Atg19.”

5. The sections that re-state results of Su et al. could be eliminated or shortened.

In reference to point 1, we feel that presenting the X-ray and EM structure is still important for reasons of validation. Furthermore, we believe that these sections have not been over-emphasized, as can be judged by the small proportion of the manuscript dedicated to this.

Specific points

6. The authors need to be more clear about what they are using as a reference for quantitation of molecular copy number. Presumably they used Nuf2, but this itself is calibrated using Cse4,

the quantification of which shows some variability from paper to paper.

Although the procedure has been published previously in more depth (Picco et al. 2015), we realize we should be more exact about the details of the procedure. Therefore, we added the following description to the revised Methods section of the manuscript on page 23:

“...the number of prApe1-GFP, Ams1-GFP or Atg19-GFP molecules was quantified using Nuf2 molecules as a reference. For each kinetochore patch 280.6 ± 16.1 molecules of Nuf2 were counted. The number of Nuf2 molecules was quantified by comparing Nuf2 and Cse4 and by considering 5 molecules of Cse4 for kinetochore (Lawrimore et al., 2011, Picco et al., 2015). The image stacks were background corrected by subtraction of the median filtered image to the image itself. The fluorescence intensity of the patches was measured by integrating the fluorescence intensity of each patch through the frames of the stack. Patches at the beginning or at the end of the stack were discarded. For Ams1, Ape1 and Atg19 only the patches that were colocalizing with an mCherry marker were quantified: Atg19-mCherry for Ams1 or Ape1 and Ape1-mCherry for Atg19.”

7. The validation of the EM reconstructions and the documentation provided for review are not up to current standards. Tilt pair analysis, particle orientation distributions, and model correlation coefficients need to be shown. The maps and models need to be provided for review. The validation of the EM reconstructions and the documentation provided for review are not up to current standards.

As requested for completion, we have added the orientation distributions of the two EM reconstructions to the revised version of the manuscript (Figures EV1E and EV3A). Thereafter, Reviewer #1 voices potential EM reconstruction concerns that at some point have been raised in the EM community but that do not apply to the presented structures. For clarification, the solved structures are of such quality that all the suggested tests do not provide meaningful support to the correctness of the structures. As already presented in our submitted paper (Figure 1G and Figure 2E-K of the manuscript), the most convincing test is always the agreement and fitting of existing high-resolution structures and/or homology models into the EM map. For mApe1, the low-resolution structure very closely resembles the simulated map of our crystal structure, a comparison which we have now added to Figure EV2F. For Ams1, the 6.3 Å resolution density map as it is represented in Figure 2 of the manuscript contains very clearly defined secondary structure features at the expected resolution. In order to improve the presentation, we have added a 3D molecular movie with the cryo-EM structure including the fitted atomic coordinates and submitted it alongside the manuscript for inspection (Movie EV1).

The proposed test of tilt-pair analysis is relevant only when the density cannot be validated by high-resolution structures. In addition, when random conical tilt was employed as in our case, the tilt-pair test is not required as the initial model was reconstructed based on angular restraints derived from tilting the stage. Finally, model correlation coefficients are rarely reported for EM structures as they are completely feature dependent and such values bear no absolute meaning in terms of model validation. In conclusion, the structural fits and comparison with atomic models are more meaningful than any of the suggested tests. We also uploaded the maps and atomic models for inspection to the referee.

The manuscript by Bertipaglia et al. reports the structural characterization of autophagic cargo - receptor complexes in order to obtain insights into the process of selective autophagy. To this end the authors have focused on the *S. cerevisiae* cytoplasm-to-vacuole-transport (Cvt) pathway during which the oligomeric prApe1 protease as well as the glycosidase Ams1 are transported into the vacuolar lumen. This transport requires the Atg19 cargo receptor, which binds the cargo proteins and links them to the membrane of the Cvt vesicles. prApe1 is initially synthesized with a N-terminal pro-peptide that is cleaved off in the lumen of the vacuole to generate the active mApe1 protease. Although the Cvt pathway is a specialized pathway in *S. cerevisiae* it has served as model pathway to uncover factors and principles underlying selective autophagy in general. In fact many genes required for autophagy also in higher eukaryotes were identified using the Cvt pathway as model. Furthermore, the concept of cargo receptors was first introduced by analysis of this pathway.

Bertipaglia et al. determined the structure of the *S. cerevisiae* prApe1 particle using negative stain EM and also determine the crystal structure of mApe1 from *C. thermophilum* using x-ray crystallography. The crystal structure of the *S. cerevisiae* mApe1 was recently published by another group (Su, 2015, Autophagy). The structure of the Ams1 cargo determined by cryo-EM is also presented. The authors then show that the Atg19 cargo receptor directly binds

to prApe1 particles (it was already known that it binds to the prApe1 pro-peptide (Scott et al., 2011, Mol Cell; Sawa-Makarska et al., 2014, NCB).

Interestingly the presence of Atg19 appears to restrict the size of the prApe1 particles suggesting that Atg19 might limit the size of the Cvt particles *in vivo*.

Employing quantitative fluorescence microscopy the authors went on to determine the number of prApe1, Ams1 and Atg19 molecules in Cvt vesicles. Finally, the authors use correlative light and electron microscopy to reveal intermediates during Cvt vesicle formation.

In summary the manuscript represents an interesting and thorough structural investigation of Cvt particle/vesicle formation. It will be of interest for scientists working in the field of selective autophagy or membrane dynamics.

We are grateful for the positive assessment of the article.

Specific comments

1a) One of the shortcomings of the manuscript is that the models derived from the structural data were not tested *in vivo*. Most of the structural data may not be of high enough resolution to introduce targeted mutations.

As previously stated in response to Reviewer #1 point 4, we have identified a point mutant of Ams1, which predominantly forms dimers. Furthermore, we have now added strong *in vivo* experiments to the structural observation that Atg19 solubilizes chain-forming prApe1 aggregates (point below). Structure-based mutations of mApe1 have been dealt with in Su et al.

1b) However, the authors propose that Atg19 limits the size of the prApe1 particles. Thus, does overexpression of Atg19 result in smaller Cvt particles? This could be quantified using the method employed for Figure 4.

In order to test our hypothesis we knocked out Atg19 in yeast and observed prApe1 puncta greater than two times brighter when compared with those seen in wildtype cells. The additional data are now shown in additional panels of Figure 3 (Figures 3H and I). We have included the following paragraph on page 9 of the revised version of the manuscript:

“Given our observation, we hypothesized that prApe1 forms larger structures in the absence of Atg19 *in vivo*. In order to test this hypothesis, we labeled Cvt cargo prApe1 with green-fluorescent protein (GFP) in *S. cerevisiae* ypt7Δ strains, which led to enrichment of prApe1 puncta due to block of vesicle fusion with the vacuole (Kim et al., 1999). Subsequently, we analyzed the intensity of prApe1-GFP spots in living cells, either in the presence or in the absence of Atg19 (Figures 3H and EV4E). In line with our hypothesis, we measured the prApe1 fluorescence intensity in the Atg19 knockout strain to be on average 2.3 ± 0.6 times brighter when compared with the wild type control strain (Figure 3I).”

2a) In Figure 3D a SEC run of the prApe1 - Atg19 complex is shown. The profiles of the individual proteins should also be shown.

As requested, we have included the control SEC profiles of the individual proteins Atg19 and prApe1 (Figures EV4A and EV4B).

2b) Also, could the authors use this method to show that the prApe1 particles become smaller in the presence of Atg19?

In the suggested case, we will need to discriminate molecular masses of >1.4 MDa (2 x prApe1) assemblies and a 730 – 830 kDa (prApe1/Atg19) complex. Common size-exclusion chromatography (SEC) columns (e.g. Superose 6) do not separate complexes beyond 1 MDa and show poor resolution in the required range of ~700 kDa and >1 MDa. In addition, we observed a non-ideal migration behavior of the assemblies with respect to commonly used size standards presumably due to their non-globular shape. Assuming co-existence of both complexes at equilibrium conditions, the limited size resolution of the column would lead to convoluted MALS profiles and hence prohibit identification of either type of complex by absolute scattering. Therefore, we concluded that SEC is not the best method to show the solubilization effect of Atg19 on prApe1 assemblies. Instead, we present evidence from the pelletation/co-sedimentation assay (Figure EV4C) and negative staining EM (Figure 3E) of the manuscript and believe the results are very convincing and more

straightforward to evaluate especially in conjunction with the new in vivo data presented in response to point 1b above.

3. Figure 3E-F show the negative stain EM analysis of prApe1 - Atg19 complexes. The quantification strongly suggests that Atg19 is preferentially associated with smaller (i.e. single prApe1 particles). Surprisingly, although the prApe1 - Atg19 is very stable (Figures 3C, D) only 20% of the prApe1 particles show density for Atg19. Where does this discrepancy derive from?

Our MBP-Atg19 pull-down and SEC data show that the prApe1/Atg19 complex can be formed in vitro whereas in negative staining EM only 20 % of prApe1 show Atg19 bound. The biochemical methods utilized allow no quantification of the affinity of complex formation, in particular its off-rate. In negative staining EM, the complex solution is dried down and as a result it is often observed that complexes “fall apart” due to the complex disruption at the air-water interface and the interactions with the carbon support film. We think the observed discrepancy is due to such effects resulting from EM grid preparation.

4. In Figure 3B the authors show the bead fraction of a MBP pull down assay. The input fraction should also be shown.

As requested by Reviewer #2, we now present the input fraction next to the bead fraction of the pull-down assay in Figure 3B.

5. The authors write in the introduction "Currently, more than 35 proteins have been shown to be involved in this process (Mizushima et al., 2011)". However, to my knowledge there are currently 41 Atg proteins (Yao et al., 2015, Autophagy).

We updated the latest number from the Yao et al. paper.

6. On page 10 the authors refer to Figure S5 instead of Figure S4.

We thank the Reviewer #2 for identification of this error, which will be corrected in the revised manuscript.

Summary of major changes

Change	Affected items
1. Addition of orientation distribution mApe1 EM structure	Added panel to Figure EV1 (EV1E)
2. Addition of structure comparison: mApe1 simulated vs. experimental EM density	Added panel to Figure EV2 (EV2F) and text on page 6
3. Addition of orientation distribution Ams1 EM structure	Added panel to Figure EV3 (EV3A)
4. Addition of 3D molecular movie of Ams1 cryo-EM structure	Added Movie EV1
5. Addition of activity assay, EM data and Atg19 binding assay on W234E Ams1 mutant	Added three panels: Figure 2L, EV3C, EV3D, EV3E and paragraph on page 7
6. Addition of Atg19's size-regulating properties on prApe1 aggregates in vivo	Added three panels: Figure 3H, 3I, Figure EV4E and paragraph on page 9
7. Addition of Atg19 and prApe1 size-exclusion chromatograms	Added two panels to Figure EV4 (EV4A/B)
8. Addition of detailed description on fluorescence quantification	Added paragraph to Methods section page 23

2nd Editorial Decision

20 April 2016

Thank you for the submission of your revised manuscript to our editorial offices. We have now received the enclosed reports on it. As you will see, the two referees find the manuscript suitable for publication in EMBO reports. Referee #1 suggests to add an additional panel to Fig. 3 (anti-Ape1 Western blot), which I suggest to do, in case you have the data. I would therefore like to ask you for further minor revisions (see below), before we can proceed with the formal acceptance of your

manuscript.

The manuscript has presently nearly 76500 words (including M+M and the references). Especially the material + methods part is extensive. Could the manuscript, in particular the methods section be shortened? Or would you consider the suggestion of referee #2 to remove part of the data?

Further, we have some requests and questions for the figures:

Fig. 1D/E - The scale bars are hard to read. Please change them to white. It would be sufficient to mention the size in the figure legend (this applies to all scale bars).

Fig. 2A/L - Should there be a third insert in A (there are three particles marked, but only two shown)? Should the selected particles be marked with circles in L? Or do we need inset boxes here?

Fig. 3D - It seems the panel in 3D shows two gels. Please separate them between lanes 14/15, or add a line and mention this in the figure legend.

Fig. 6E/F/G - The scale bars are hard to read. Please change to white.

Fig. EV1A/B and EV4B - Please also separate or mark by a line, if panels were assembled from different gels (as Fig. 3D) and mention this in the figure legends.

Fig. EV5 - The scale bars are hard to read. Please change to white.

We also could not find any reference or legend for the four datasets. What are these?

I look forward to seeing a revised version of your manuscript when it is ready. Please let me know if you have questions or comments regarding the revision.

REFEREE REPORTS

Referee #1:

The authors have addressed all my comments satisfactorily. Perhaps the authors could add an anti-Ape1 western blot to Figure 3H,I/ EV4E to show that the levels of this protein are similar in the two yeast strains used.

Referee #2:

In general the authors have done an excellent job on this revision. This is superb work and the further characterization of the Ams1 interfaces and other improvements strengthen the paper.

The manuscript seems quite long by the standards of EMBO Reports at 11 figures. I still think the portions that essentially repeat Su et al., although proportionately reduced from the original version, are not really necessary to publish in this venue. The authors outline some minor differences between Su et al. and their results (*C. thermophilum* here, *S. cerevisiae* in Su et al, for example) but I am still not persuaded that these differences amount to anything that will be of interest to the autophagy community broadly. This part could be excised and published in a structural specialty journal if space is tight.

2nd Revision - authors' response

29 April 2016

We have prepared a revised version of the manuscript according to your requests (find detailed point-by-point response below). Briefly, the revised manuscript has been shortened significantly and contains the requested prApe1 western blot of the *in vivo* Atg19 size control experiments.

With the new version of the revised manuscript, we hope that we can convince you that the novelty and importance of our work justifies publication in *EMBO Reports* as a Research Article.

The manuscript has presently nearly 76500 words (including M+M and the references). Especially the material + methods part is extensive. Could the manuscript, in particular the methods section be shortened? Or would you consider the suggestion of referee #2 to remove part of the data?

We have shortened the manuscript considerably and as a result the main text now complies with the 25,000 character range. In addition, we shortened the Material and Methods section. The entire document now contains 61,700 characters (including abstract, keywords, two-sentence summary, three bullet points of highlights, main text, references, figure legends). After discussion with the co-authors, we have decided against the proposal of Reviewer #2 to take out the mApe1 crystal structure.

Further, we have some requests and questions for the figures:

Fig. 1D/E - The scale bars are hard to read. Please change them to white. It would be sufficient to mention the size in the figure legend (this applies to all scale bars).

Changed as suggested.

Fig. 2A/L - Should there be a third insert in A (there are three particles marked, but only two shown)? Should the selected particles be marked with circles in L? Or do we need inset boxes here?

(A) This point may have arisen from a misunderstanding of the type of data. We removed the circles around the particles as they did not correspond to the displayed class averages. Therefore, we have moved the row of particles below the micrograph as they represent class averages that are a result of averaging of particles from many micrographs.

(L) Similarly the class averages do not present selected particles on the micrograph. They are the result of an independent averaging process.

Fig. 3D - It seems the panel in 3D shows two gels. Please separate them between lanes 14/15, or add a line and mention this in the figure legend.

Changed as requested.

Fig. 6E/F/G - The scale bars are hard to read. Please change to white.

Changed as requested.

Fig. EV1A/B and EV4B - Please also separate or mark by a line, if panels were assembled from different gels (as Fig. 3D) and mention this in the figure legends.

Changed as requested.

Fig. EV5 - The scale bars are hard to read. Please change to white.

Changed as requested.

We also could not find any reference or legend for the four datasets. What are these?

We provided the datasets due to the specific request of Reviewer #1. The data sets have been deposited to the EM Data Bank and the Protein Data Bank and will not be distributed as part of the manuscript. At the bottom of the manuscript, we mention the newly obtained accession IDs.

I am very pleased to accept your manuscript for publication in the next available issue of EMBO reports. Thank you for your contribution to our journal.

Corresponding Author Name: Carsten Sachse
Journal Submitted to: EMBO reports
Manuscript Number: EMBOR-2015-41960V3